# Precipitation of Mn Oxides in Quaternary Microbially Induced Sedimentary Structures (MISS), Cape Vani Paleo-Hydrothermal Vent Field, Milos, Greece

**Stephanos P. Kilias [1,\*]**, **Magnus Ivarsson [2,3]**, **Ernest Chi Fru [4]**, **Jayne E. Rattray [5,†]**, **Håkan Gustafsson [6]**, **Jonathan Naden [7]** and **Kleopatra Detsi [1]**

[1] Department of Economic Geology and Geochemistry, Faculty of Geology and Geoenvironment, National and Kapodistrian University of Athens, Panepistimiopolis, Zographou, 15784 Athens, Greece; detsi@geol.uoa.gr

[2] Department of Biology, University of Southern Denmark, Campusvej 55, 5230 Odense, Denmark; Magnus.Ivarsson@nrm.se

[3] Department of Paleobiology, Swedish Museum of Natural History, Box 50007, 10405 Stockholm, Sweden

[4] School of Earth and Ocean Sciences, Centre for Geobiology and Geochemistry, Cardiff University, Park Place, Cardiff CF10 3AT, UK; chifrue@cardiff.ac.uk

[5] Department of Geological Sciences, Stockholm University, SE-106 91 Stockholm, Sweden; jayne.rattray@ucalgary.ca

[6] Center for Medical Image Science and Visualization (CMIV), Linköping University, 581 83 Linköping, Sweden; hakan.l.gustafsson@liu.se

[7] British Geological Survey, Keyworth, Nottingham NG12 5GG, UK; jna@nigl.nerc.ac.uk

\* Correspondence: kilias@geol.uoa.gr

† Present address: Department of Biological Sciences, University of Calgary, Calgary, AB T2N 1N4, Canada.

**Abstract:** Understanding microbial mediation in sediment-hosted Mn deposition has gained importance in low-temperature ore genesis research. Here we report Mn oxide ores dominated by todorokite, vernadite, hollandite, and manjiroite, which cement Quaternary microbially induced sedimentary structures (MISS) developed along bedding planes of shallow-marine to tidal-flat volcaniclastic sandstones/sandy tuffs, Cape Vani paleo-hydrothermal vent field, Milos, Greece. This work aims to decipher the link between biological Mn oxide formation, low-T hydrothermalism, and, growth and preservation of Mn-bearing MISS (MnMISS). Geobiological processes, identified by microtexture petrography, scanning and transmission electron microscopy, lipid biomarkers, bulk- and lipid-specific $\delta^{13}C_{organic}$ composition, and field data, and, low-temperature hydrothermal venting of aqueous $Mn^{2+}$ in sunlit shallow waters, cooperatively enabled microbially-mediated Mn (II) oxidation and biomineralization. The MnMISS biomarker content and $\delta^{13}C_{org}$ signatures strongly resemble those of modern Mn-rich hydrothermal sediments, Milos coast. Biogenic and syngenetic Mn oxide precipitation established by electron paramagnetic resonance (EPR) spectroscopy and petrography, combined with hydrothermal fluid flow-induced pre-burial curing/diagenesis, may account for today's crystalline Mn oxide resource. Our data suggests that MISS are not unique to cyanobacteria mats. Furthermore, microbial mats inhabited by aerobic methanotrophs may have contributed significantly to the formation of the MnMISS, thus widening the spectrum of environments responsible for marine Mn biometallogenesis.

**Keywords:** MISS; Milos; hydrothermal Mn oxide; lipid biomarkers; lipid specific stable carbon isotopes; electron paramagnetic resonance (EPR) spectroscopy

## 1. Introduction

The link between geobiology and economic geology, i.e., the role of microbial activity on the formation of mineral ore deposits, is a relatively new topic [1–3]. One of the best examples is the microbial precipitation of Mn minerals and the formation of economic sediment-hosted Mn ore deposits [2,4–6]. Due to the high redox potential of Mn, its enrichment in both modern and ancient sediment-hosted deposits reflects multiple reversal redox conversions {Mn(II)↔Mn(III,IV)} [4,5,7]. Abiotic $Mn^{2+}_{(aq)}$ oxidation can occur in natural systems: (1) via superoxide ($O_2^-$), generated by sunlight-driven nitrate photolysis [8]; (2) with elevated pH [9]; (3) via autocatalysis on mineral surfaces [10,11]; and, (4) via photochemical processes [12,13]. However, the accepted pathway of concentration of Mn minerals in the marine sedimentary rock record involves direct processing by diverse Mn(II)-oxidising and Mn(III/IV)-reducing bacteria and fungi [4,5,7,14,15], or by the, indirect activities of microbial processing by phototrophic green algae, diatoms, and cyanobacteria [16].

Biological oxidation of dissolved Mn(II) species to solid phase Mn(III,IV) (oxyhydr)oxides (hereafter Mn oxides) may involve: (1) enzymatic pathways, which implicate multicopper Mn oxidases in some bacteria and fungi [4,17–20]; (2) Mn-oxidizing peroxidase (or MopA) by various bacteria [21] and fungi [5,22,23]; (3) superoxide ($O_2^-$) of fungal, bacterial, or phototrophic microbial origin [15,16,24,25]; and (4) anoxygenic photosynthetic microbial biomineralization of Mn (in the presence of sulphides) in the photic zones of modern anoxic sediments [26]. Oxidation of Mn(II) may also involve interdependent enzymatic and reactive-Mn oxide surface-induced pathways accelerated in the presence of bacterial metabolites and sunlight [12].

Modern Fe-Mn crusts, Mn nodules and, sediment-hosted hydrothermal Mn oxides are dominated by the Mn(IV) oxides, birnessite [(Na,Ca)$Mn_7O_{14}\cdot 2.8H_2O$], todorokite [(Ca,Na,K) × ($Mn^{4+},Mn^{3+})_6O_{12}\cdot 3.5H_2O$], and vernadite ($\delta$-$MnO_2$) [27,28], whereas kutnohorite [$CaMn(CO_3)_2$], rhodochrosite ($MnCO_3$), and braunite ($Mn(III)_6Mn(II)O_8SiO_4$) dominate ancient deposits [6,29]. Most recent publications on Mn biometallogenesis deal with Mn-carbonate (±Mn-silicate, Mn oxide) deposits ranging in age from Precambrian to Mesozoic, and the role microbes play in their genesis [2,3,6,30–38]. Manganese enrichment in black shale-hosted Mn-carbonate deposits is thought to start with microbially mediated oxidation of hydrothermally sourced $Mn^{2+}_{(aq)}$ to solid $Mn^{3+/4+}$ oxide proto-ore, subsequently reduced by microbial heterotrophic oxidation of organic carbon during early diagenesis resulting in Mn carbonate ore [2,3,5–7,29,35]. A biogeochemical origin has been assigned to the Cretaceous sediment-hosted Groote Eylandt deposit, which has both carbonate- and oxide-dominated orebodies [39,40].

Despite the inference that Mn oxide formation is largely a biomineralization process, identifying geobiological evidence in the marine sedimentary record that are conducive and favorable for the formation of Mn oxide ores under conditions relevant to low-temperature hydrothermalism remains an unsettled area of Mn biometallogenesis research [2,3,29]. Here we report potentially biogenic hydrothermal Mn oxide mineralization hosted by microbially induced sedimentary structures (MISS) in a paleohydrothermal vent field associated with Mn-ore on Milos Island, Greece. We combine results from petrographic microtexture analyses, scanning electron microscopy (SEM) and transmission electron microscopy (TEM), organic (lipid) analyses and bulk and lipid specific stable $C_{organic}$ isotope composition of matter, with electron paramagnetic resonance (EPR) spectroscopy, as well as previous field and fluid inclusion data. We demonstrate that when geothermal metalliferous fluids vent into the shallow euphotic environments, their interplay with microbial mat colonization, physical siliciclastic sediment dynamics, and pre-burial curing/diagenesis, result in a mineralizing environment that favours biologically mediated $Mn^{2+}_{(aq)}$ oxidation, and plays a significant role in Mn fixation in Mn(III/IV) oxides. The Cape Vani Mn mineralization may serve as a 'biomarker' for the former presence of microbial mats, and widens the spectrum of conditions and geobiological processes responsible for marine Mn–ore formation to include Quaternary sunlit shallow-marine τo tidal-flat siliciclastic sedimentary environments associated with low-temperature hydrothermal fluid flow.

*Geology, Mn Mineralization and "Microbially Induced Sedimentary Structures" (MISS) of the Cape Vani Mine*

The geology and paleo-hydrothermal setting of the Fe and Mn mineralization of the Cape Vani Sedimentary basin (CVSB), have previously been described in detail [41–51], and are briefly presented here.

The Cape Vani Mn deposit is located in the NW extremity of Milos Island, a recently emergent <2 Ma volcano of the active Hellenic volcanic arc (HVA) (Figure 1a) that is characterized by volcanism and geothermal activity through thinned pre-Alpine to Quaternary continental crust [52]. Milos comprises a calc-alkaline volcanic-sedimentary rock succession that has evolved from Middle via Late Pliocene (3.5–2.7 Ma) to Late Pleistocene (<0.08 Ma), and records a transition from a relatively shallow submarine to subaerial settings [45,53]. Extensional tectonics resulted in a series of horsts and grabens that controlled volcanic and hydrothermal activity [54]. Milos currently hosts an active high-enthalpy geothermal system (i.e., conventionally exploitable at an average of 3–5 MW electric power per well) with expressions in the shallow-marine to subaerial environment [55–57], which has produced Late Pliocene to Early Pleistocene submarine, and transitional volcanogenic massive sulphide (VMS) and epithermal, mineralization [46,58–60].

Cape Vani operated as a Mn mine from 1886 to 1909 and from 1916 to 1928 producing 220,000 tons of Mn [61]. Recently, iron formations have been discovered in the CVSB, which display banded rhythmicity similar to Precambrian banded iron formation (BIF), and are considered Earth's youngest Precambrian BIF analogues of Quaternary age [49–51]. This paper describes the economic Mn oxide ore deposit, which contains an inferred resource of 2.1 million tons with mean composition of 14.6 wt.% Mn, 12.8 wt.% $BaSO_4$, 62 wt.% $SiO_2$, and 7.5 wt.% $Fe_2O_3$ [42,43]. Whole rock geochemical data shows that the Mn-ores are enriched in K, Na, Mg, Ca, Al, Ti, Fe, Zr, Nb, Ce, Hf, and Th, and, Ba, Pb, Zn, As, Sb, and W metal, whereas rare earth elements (REE) patterns are characterized by negative Ce anomalies and positive Eu anomalies, which characterize marine hydrothermal Mn deposits, and, the presence of K-feldspar and barite, respectively [43]. Manganese minerals include vernadite ($\delta$-$MnO_2$), pyrolusite, ramsdellite, cryptomelane, hollandite, coronadite, and lesser romancheite, jacobsite, franklinite and hydrohetaerolite [41–43,46,62].

The CVSB, host of the Mn and Fe ores, covers ~1 $km^2$ area and is a restricted Quaternary intravolcanic rift basin, that is floored by dacitic–andesitic lava domes and filled with >60 m thick Upper Pliocene to Lower Pleistocene fossiliferous volcaniclastic/siliciclastic sandstones/sandy tuffs. These sediments record a shallow-marine to shoreline depositional setting characterized by tidal influence. Over the last 0.8 Myr, fluctuating water depths due to a sea level change of up to 120 m, and volcanic edifice building, have resulted in tectonic uplift of approximately 250 m [54]; this largely precludes any substantial (~1 km) burial, suggesting that there has been little or no burial diagenesis of sandstones/sandy tuffs. The CVSB is divided into three lithologically variable, fault-bounded volcano-sedimentary sub-basins [50] (Figure 1b): (A) Basin 1, hosts microfossil-rich banded iron formation (BIF)-type deposits (microfossiliferous iron formation, MFIF), as well as conglomerate-hosted iron formation (CIF), and small part of the bulk Mn oxide ores; (B) Basin 2, host of the bulk ore-grade Mn oxide deposit, and, (C) Basin 3, host of non-fossiliferous BIF-type deposits (non-fossiliferous iron formation, NFIF) intercalated with Mn oxide ores. The CVSB hosts a fossil analogue of active shallow submarine hydrothermal activity on the Milos coast, featured by quartz-barite (Mn oxide, Fe oxide, chalcedony, K-feldspar) hydrothermal feeder veins, which cut the entire volcano-sedimentary section, and pervasive hydrothermal alteration of the siliciclastic sediments (barite, silica, K-feldspar, illite) [41–43,46,62].

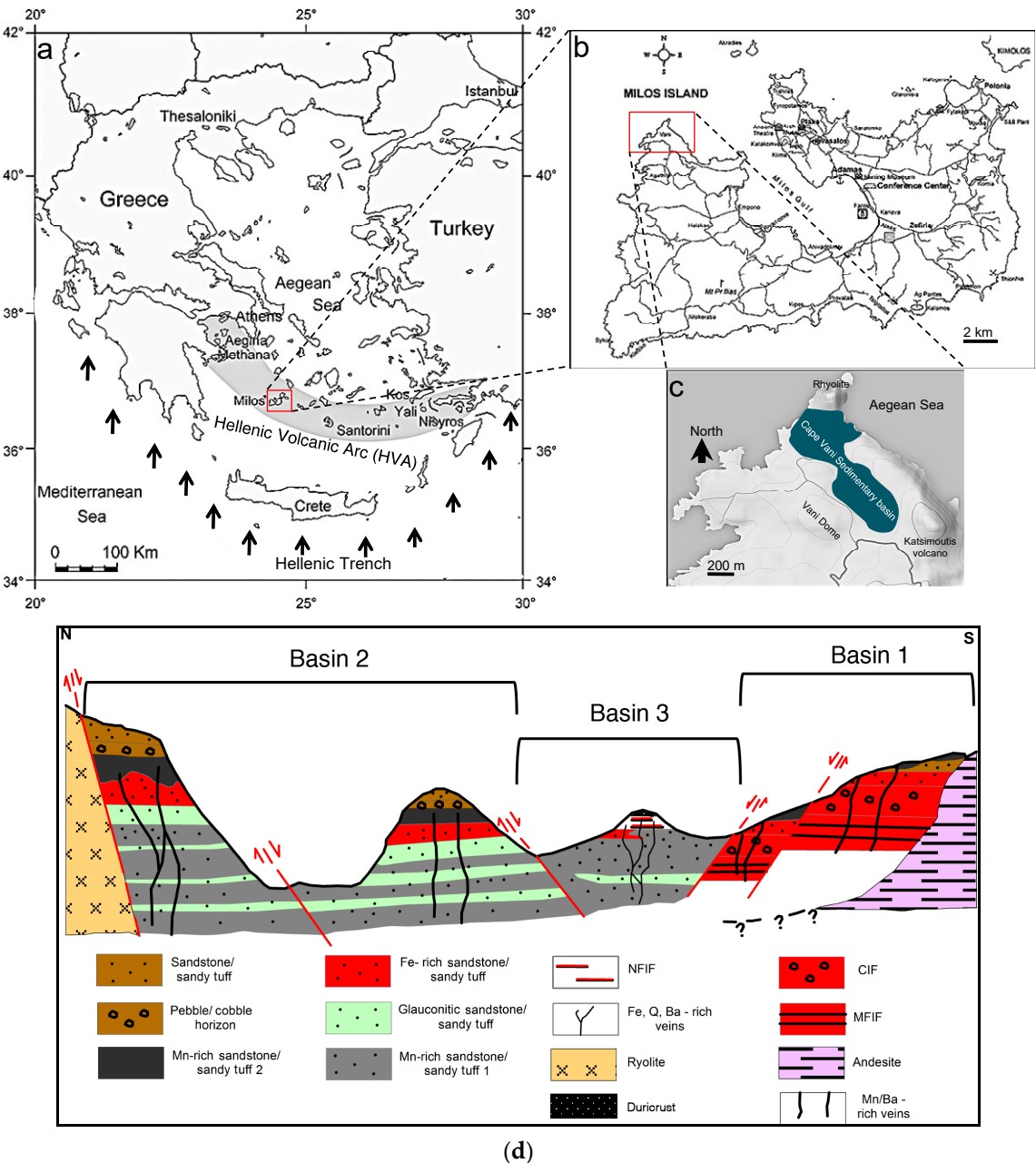

**Figure 1.** (**a**) The location of Milos Island in the Aegean Sea, showing the Hellenic Volcanic Arc (HVA). Arrows indicate northeast subduction of the African plate underneath the Eurasian plate; (**b**) Milos Island; (**c**) the eight-shaped Cape Vani sedimentary basin (CVSB); (**d**) a generalized north-south stratigraphic map of the ~1 km long Cape Vani sedimentary basin (CVSB) showing interpreted geology, lithology, and main faults. CIF: conglomerate-hosted iron formation; NFIF: non-fossiliferous iron formation; MFIF: microfossiliferous iron formation [50,51].

The Mn oxide ore exhibits a range of deposit varieties, which include an abundant and diverse array of microbial mat-related structures (MRS), which are heterogeneously distributed in the host sandstone/sandy tuff and the CVSB; moreover, they are closely associated with the quartz-barite feeder veins. Based on biogenicity criteria, these MRS have been identified as Mn–mineralized microbially induced sedimentary structures (MnMISS) formed due to the response of microbial mats to physical sediment dynamics [63,64]. This interpretation is strengthened by new textural, mineralogical and geochemical data (this paper). Moreover, the Cape Vani MnMISS are closely associated with

the quartz-barite feeder veins, signifying the link between MISS growth and low temperature seafloor hydrothermal fluid venting.

Field relationships, combined with fluid inclusion data from hydrothermal barite and quartz, suggest that the MnMISS developed in a very shallow (<50 m) steaming marine paleo-environment, in close association with white smoker-type seafloor activity that probably acted as Mn(II) supplier. The temperatures were ~125 °C at depth (feeder veins) cooling to <100 °C near and at the paleo-seafloor in response to boiling, mixing with seawater and conductive heat loss [43,62]. Sulphur ($\delta^{34}$S) and oxygen ($\delta^{18}$O) isotope evidence has shown that the investigated MnMISS are younger than 1.5 Ma, and possibly formed about 0.5 Ma ago [43]; moreover, the MnMISS associated barite was derived from seawater sulphate that has undergone some alteration through bacterial sulphate reduction (BSR) [43]. Furthermore, fungal remains, exceptionally preserved as fluid inclusions, have been identified in hydrothermal quartz from feeder veins [43].

Selected hydrothermal features, Mn ore styles, and Mn mineralized microbially induced sedimentary structures (MISS) are shown in Supplementary Figure S1.

## 2. Materials and Methods

### 2.1. Sampling

Samples used in this study were mostly collected from Basin 2 outcrops (Figure 1b) inside the abandoned Cape Vani Mn mine. The geological and stratigraphic characteristics, morphological variations and outcrop diversity of the samples have been detailed in previous studies [46,50,62]. Selected sampling sites with sample numbers are shown in Supplementary Figure S1. For the needs of the EPR study, complementary samples were collected from feeder stockwork feeder veins stratigraphically below the mine. Sample numbers and corresponding MISS type [46], whole rock geochemistry, and mineralogical composition of the analyzed samples can also be found in Table 1 and Supplementary Table S1 (see below).

**Table 1.** Whole-rock chemical analyses of selected major and trace elements in different Mn-mineralized microbially induced sedimentary structures (MnMISS).

| Sample | MISS Type [1] | Fe | Ca | P | Mg | Ti | Al | K | Na | S | SiO$_2$ | Mn | |
|---|---|---|---|---|---|---|---|---|---|---|---|---|---|
| | | | | | | wt.% | | | | | | | |
| MI-10–28 | upturned margins | 0.61 | 0.18 | 0.030 | 0.15 | 0.127 | 5.32 | 3.86 | 0.134 | <0.1 | 42.69 | 7.22 | |
| MI-10–37 | Mn nodules | 7.98 | 0.12 | 0.031 | 0.10 | 0.082 | 3.18 | 2.96 | 0.100 | <0.1 | 26.31 | 14.18 | |
| MI-10–20 | growth bedding | 1.11 | 0.19 | 0.005 | 0.06 | 0.201 | 4.93 | 4.00 | 0.257 | <0.1 | 44.26 | 14.88 | |
| MI-10–24 | fossil gas dome | 1.75 | 0.19 | 0.014 | 0.28 | 0.111 | 4.48 | 4.22 | 0.219 | 0.1 | 36.09 | 14.62 | |
| MI-10–26 | mat layer structure | 3.22 | 0.15 | 0.006 | 0.03 | 0.043 | 2.07 | 3.17 | 0.060 | <0.1 | 29.12 | 10.71 | |
| MI-10–29 | Mn nodules | 1.58 | 0.13 | 0.009 | 0.11 | 0.163 | 4.11 | 2.47 | 0.094 | <0.1 | 35.86 | 14.49 | |
| MI-10–12 | roll-up structure | 3.89 | 0.34 | 0.014 | 0.24 | 0.051 | 1.94 | 2.57 | 0.286 | <0.1 | 18.41 | 24.86 | |
| MI-10–15 | roll-up structure | 2.25 | 0.43 | 0.023 | 0.29 | 0.038 | 2.28 | 3.58 | 0.442 | <0.1 | 30.12 | 29.58 | |
| MI-10–22 | mat fragments/chips | 1.71 | 0.09 | 0.005 | 0.09 | 0.056 | 4.24 | 3.73 | 0.161 | <0.1 | 33.47 | 13.29 | |
| MI-10–27 | upturned margins | 0.67 | 0.10 | 0.032 | 0.06 | 0.132 | 4.96 | 3.10 | 0.097 | <0.1 | 38.9 | 9.80 | |
| MI-10–26S [2] | sandstone/sandy tuff | 2.52 | 0.11 | 0.004 | 0.05 | 0.101 | 3.92 | 5.22 | 0.087 | 0.2 | 62.05 | 0.92 | |
| MI-10–29S [2] | sandstone/sandy tuff | 1.14 | 0.06 | 0.003 | 0.25 | 0.252 | 6.70 | 6.41 | 0.089 | 0.2 | 64.7 | 0.95 | |
| MI-27S [2] | sandstone/sandy tuff | 0.67 | 0.11 | 0.031 | 0.06 | 0.183 | 6.56 | 5.52 | 0.097 | 0.1 | 64.13 | 0.66 | |
| MI-34S [2] | sandstone/sandy tuff | 0.68 | 0.07 | 0.006 | 0.02 | 0.064 | 2.42 | 3.62 | 0.062 | <0.1 | 72.33 | 3.07 | |
| MI-34 | wrinkle structures | 2.66 | 0.15 | 0.015 | 0.06 | 0.055 | 2.78 | 3.88 | 0.082 | <0.1 | 33.03 | 9.63 | |
| MI-33 | wrinkle structures | 0.76 | 0.08 | 0.006 | 0.09 | 0.110 | 2.90 | 4.14 | 0.212 | <0.1 | 66.71 | 6.41 | |
| **Sample** | **MISS Type [1]** | **Zn** | **Ag** | **Ni** | **Co** | **As** | **U** | **Th** | **Sr** | **Cd** | **Sb** | **V** | **La** |
| | | | | | | ppm | | | | | | | |
| MI-10–28 | upturned margins | 1802 | 7.2 | 4.2 | 8.1 | 736 | 4.0 | 7.1 | 592 | 1.6 | 43.0 | 11 | 11.7 |
| MI-10–37 | Mn nodules | 6881 | 7.2 | 9.1 | 78.6 | 3551 | 6.5 | 3.1 | 449 | 13.0 | 126.8 | 21 | 16.2 |
| MI-10–20 | growth bedding | 5216 | 35.8 | 9.9 | 35.6 | 2020 | 7.6 | 4.1 | 1652 | 5.3 | 293.1 | 91 | 23.2 |
| MI-10–24 | fossil gas dome | 6884 | 9.4 | 4.5 | 4.2 | 511 | 2.5 | 5.1 | 638 | 23.8 | 45.7 | 11 | 10.9 |
| MI-10–26 | mat layer structure | 3679 | 12.8 | 3.4 | 9.4 | 2031 | 4.5 | 1.4 | 2232 | 2.1 | 235.5 | 30 | 16.9 |
| MI-10–29 | Mn nodules | 2632 | 24.6 | 8.6 | 15.8 | 1457 | 7.4 | 5.3 | 1336 | 4.0 | 161.6 | 56 | 16.5 |
| MI-10–12 | roll-up structure | 4225 | 26.0 | 13.6 | 12.0 | 302 | 5.4 | 2.2 | 2125 | 40.9 | 8.1 | 22 | 27.0 |
| MI-10–15 | roll-up structure | 4183 | 14.1 | 7.1 | 20.7 | 317 | 9.0 | 2.1 | 1782 | 37.0 | 43.1 | 1 | 22.2 |
| MI-10–22 | mat fragments/chips | 4936 | 16.3 | 7.9 | 13.0 | 2003 | 4.5 | 3.0 | 495 | 2.1 | 415.9 | 68 | 21.2 |
| MI-10–27 | upturned margins | 1458 | 9.5 | 3.6 | 8.7 | 1445 | 4.0 | 5.3 | 646 | 2.6 | 30.6 | 1 | 12.7 |
| MI-10–26S [2] | sandstone/sandy tuff | 562 | 34.4 | 2.7 | 1.6 | 618 | 2.6 | 2.5 | 2463 | 0.6 | 109.0 | 29 | 3.9 |
| MI-10–29S [2] | sandstone/sandy tuff | 495 | 43.9 | 3.4 | 2.5 | 74 | 2.9 | 5.8 | 938 | 0.2 | 33.8 | 30 | 9.1 |

**Table 1.** *Cont.*

| Sample | MISS Type [1] | Zn | Ag | Ni | Co | As | U | Th | Sr | Cd | Sb | V | La |
|---|---|---|---|---|---|---|---|---|---|---|---|---|---|
| | | ppm | | | | | | | | | | | |
| MI-27S [2] | sandstone/sandy tuff | 446 | 10.8 | 2.6 | 1.4 | 144 | 3.3 | 3.9 | 594 | 0.2 | 17.2 | 31 | 4.8 |
| MI-34S [2] | sandstone/sandy tuff | 396 | 16.8 | 1.8 | 2.6 | 170 | 1.6 | 2.2 | 1968 | 1.1 | 37.7 | 10 | 4.6 |
| MI-34 | wrinkle structures | 2449 | 13.9 | 3.7 | 20.3 | 1246 | 3.0 | 2.8 | 1957 | 5.3 | 86.2 | <1 | 12.2 |
| MI-33 | wrinkle structures | 589 | 43.6 | 5.3 | 2.5 | 211 | 2.5 | 3 | 636 | 0.5 | 68.3 | 16 | 4.7 |

| Sample | MISS Type [1] | Cr | Ba | W | Zr | Ce | Sn | Th | Y | Nb | Ta | Be | Sc |
|---|---|---|---|---|---|---|---|---|---|---|---|---|---|
| | | ppm | | | | | | | | | | | |
| MI-10–28 | upturned and curled margins | 16 | 4328 | 22.3 | 56.7 | 30 | 1.3 | 7.1 | 18.8 | 4.7 | 0.4 | 7 | 9 |
| MI-10–37 | Mn nodules | 19 | >10,000 | 55.6 | 24.9 | 34 | 1.1 | 3.1 | 20.8 | 2.3 | 0.2 | 20 | 6 |
| MI-10–20 | growth bedding | 25 | >10,000 | 103.9 | 44.6 | 47 | 1.5 | 4.1 | 16.2 | 3.5 | 0.2 | 10 | 4 |
| MI-10–24 | fossil gas dome | 15 | >10,000 | 28.0 | 37.5 | 18 | 0.7 | 5.1 | 12.3 | 3.4 | 0.3 | 6 | 5 |
| MI-10–26 | mat layer structure | 13 | >10,000 | 66.7 | 12.0 | 25 | 0.8 | 1.4 | 8.0 | 1.1 | <0.1 | 12 | 2 |
| MI-10–29 | Mn nodules | 24 | >10,000 | 34.4 | 37.6 | 22 | 1.8 | 5.3 | 9.4 | 4.8 | 0.4 | 2 | 3 |
| MI-10–12 | roll-up structure | 17 | >10,000 | 4.2 | 14.2 | 45 | 0.6 | 2.2 | 59.7 | 1.5 | 0.1 | 2 | 5 |
| MI-10–15 | roll-up structure | 19 | >10,000 | 5.7 | 11.3 | 29 | 0.6 | 2.1 | 33.2 | 1.3 | <0.1 | 5 | 2 |
| MI-10–22 | mat fragments/chips | 22 | >10,000 | 157.2 | 20.8 | 18 | 1.1 | 3.0 | 9.0 | 2.0 | 0.2 | 23 | 3 |
| MI-10–27 | upturned and curled margins | 14 | >10,000 | 16.1 | 40.1 | 19 | 1.3 | 5.3 | 15.9 | 4.2 | 0.3 | 6 | 5 |
| MI-10–26S [2] | sandstone/sandy tuff | 10 | 1128 | 24.4 | 26.0 | 7 | 0.9 | 2.5 | 5.3 | 2.4 | 0.2 | 3 | 3 |
| MI-10–29S [2] | sandstone/sandy tuff | 13 | 2912 | 12.4 | 60.5 | 15 | 1.9 | 5.8 | 6.6 | 8.6 | 0.6 | 2 | 5 |
| MI-27S [2] | sandstone/sandy tuff | 13 | 4653 | 12.4 | 42.8 | 10 | 1.8 | 3.9 | 8.1 | 5.6 | 0.3 | 1 | 2 |
| MI-34S [2] | sandstone/sandy tuff | 9 | 3172 | 4.3 | 22.8 | 9 | 0.7 | 2.2 | 3.6 | 2.1 | 0.2 | 1 | >1 |
| MI-34 | wrinkle structures | 16 | >10,000 | 11.1 | 20.1 | 15 | 0.9 | 2.8 | 11.6 | 1.6 | 0.2 | 5 | 3 |
| MI-33 | wrinkle structures | 30 | >10,000 | 4.3 | 22.6 | 8 | 0.9 | 3.0 | 4.2 | 2.6 | 0.2 | 2 | 2 |

**Table 1.** *Cont.*

| Sample | MISS Type [1] | Li | Rb | Hf | Mo | Cu | Pb |
|---|---|---|---|---|---|---|---|
| | | ppm | | | | | |
| MI-10–28 | Upturned margins | 15.8 | 100.8 | 1.8 | 25.8 | 12.17 | >10,000 |
| MI-10–37 | Mn nodules | 7.0 | 88.0 | 0.7 | 188.8 | 2423 | >10,000 |
| MI-10–20 | growth bedding | 6.3 | 108.6 | 1.4 | 13.1 | 787.0 | >10,000 |
| MI-10–24 | fossil gas dome | 4.2 | 124.1 | 1.1 | 11.0 | 35.8 | 949.8 |
| MI-10–26 | mat layer structure | 6.1 | 93.0 | 0.4 | 17.7 | 484.6 | >10,000 |
| MI-10–29 | Mn nodules | 10.7 | 67.7 | 1.2 | 54.7 | 1351 | >10,000 |
| MI-10–12 | roll-up structure | 9.5 | 81.8 | 0.5 | 25.1 | 94.1 | 4561 |
| MI-10–15 | roll-up structure | 28.7 | 102.8 | 0.3 | 44.6 | 81.8 | 561.5 |
| MI-10–22 | mat fragments/chips | 6.0 | 107.3 | 0.6 | 38.3 | 828.7 | >10,000 |
| MI-10–27 | upturned and curled margins | 10.3 | 81.8 | 1.3 | 12.9 | 1244 | >10,000 |
| MI-10–26S [2] | sandstone/sandy tuff | 2.8 | 186.2 | 0.7 | 1.3 | 58.5 | 1292 |
| MI-10–29S [2] | sandstone/sandy tuff | 6.6 | 191.1 | 2.0 | 4.1 | 30.5 | 107.6 |
| MI-27S[2] | sandstone/sandy tuff | 11.3 | 183.8 | 1.3 | 1.0 | 51.0 | 2495 |
| MI-34S[2] | sandstone/sandy tuff | 22.7 | 129.5 | 0.7 | 6.0 | 72.8 | 555.3 |
| MI-34 | wrinkle structures | 23.4 | 112.2 | 0.7 | 38.2 | 357.5 | 3149 |
| MI-33 | wrinkle structures | 52.0 | 127.4 | 0.7 | 10.8 | 158.0 | 90.7 |

[1] Kilias (2012) [46], [2] denotes substrate.

## 2.2. Petrographic, Chemical and Mineralogical (Field-Emission Gun Scanning Electron Microscope (FEG-SEM), Field-Emission Gun Transmission Electron Microscope (FEG-TEM)) Analysis

Acme Analytical Labs Ltd. AcmeLabs conducted analyses on 0.2 g of whole rock samples, using 4 acid digestion inductively coupled plasma mass spectrometry (ICP-MS) analysis. Detection limits ranged from 0.001–1.0 ppm for a full range of elements. Powder X-ray diffraction (PXRD) patterns were collected using a Bruker D8 diffractometer, operating at 40 kV/40 mA and equipped with CuKa radiation (k = 1.5418) and a LynxEye detector. Samples were analysed in the 20–80° (2h) range with a 0.010 step size and step time of 155 s. Prior to SEM imaging, samples were loaded onto individual aluminium stubs and left to air dry. A platinum coating was applied to all samples prior to imaging at 20 kV using a LEO 1530 field-emission gun scanning electron microscope (FEG-SEM), at the School of Earth and Environment, University of Leeds, UK. FEG-SEM imaging was used for characterizing the morphology and distribution of Mn mineral phases in the samples. In addition, qualitative elemental analyses were carried out using an energy-dispersive spectrometer (EDS) attached to the FEG-SEM.

High-resolution TEM (HRTEM) investigations were performed on powdered samples placed onto a holey carbon support grid (Agar Scientific Ltd. Stansted, Essex, UK) using a FEI CM200 FEG-TEM operating at 197 kV with a point resolution of ca. 2.3 Å, and fitted with a Gatan Imaging filter (GIF 200), at the Institute for Materials Research, University of Leeds. Bulk energy loss spectra were taken in diffraction mode (image coupled) using a Ca. 0.18 mm diameter selected area aperture (SAD).

## 2.3. Electron Paramagnetic Resonance (EPR)

EPR spectroscopy is a technique to study paramagnetic species in an applied magnetic field, and it is used here for analysing Mn oxides to indicate a biogenic signature (see Section 4.2.1. Biogenicity"). The analysis was performed on a X-Band Bruker E500 EPR (Bruker Bio-Spin GmbH, Rheinstetten, Germany) with a 4103 TM resonator at room temperature in a clear fused quartz EPR sample tube (707-SQ-250M) from Wilmad-LabGlas (Vineland, NJ, USA). No EPR signal could be detected from the empty sample tube. Measurements were done using microwave power of 10 mW or 1 mW for comparisons, 2 G modulation amplitude, 5.12 ms time constant, 20 s sweep time (three added sweeps). Spectra were imported into MATLAB (version R2011a, MathWorks Inc., Natick, MA, USA) for analysis [65]. The EPR signal line width was determined as peak-to-peak, Δpp (the horizontal distance between the maximum and the minimum of a first-derivative line shape). The EPR spectra were further analysed by comparisons with superpositions of Gaussian lines.

## 2.4. Organic Carbon Analysis

MnMISS samples were pre-treated with concentrated HCl to remove inorganic carbon prior to $\delta^{13}C_{org}$ of organic carbon analysis and performed as previously described [49] at the Stable Isotope Laboratory at the Department of Geological Sciences, Stockholm University, Sweden. Briefly, samples were combusted in a Carlo Erba NC2500 analyzer and analyzed in a Finnigan MAT Delta V mass spectrometer, via a split interface to reduce gas volume. Reproducibility was calculated to be better than 0.15‰ for $\delta^{13}C$. Total $C_{org}$ was determined simultaneously when measuring the isotope ratios. For lipid biomarker analysis, ground rock samples (4–6 g), selected to contain areas of well-preserved Mn with limited exposure, were extracted using 3× methanol, 3× (1:1) methanol: dichloromethane (DCM), and 3× DCM. All solvents were high purity chromatography grade and glassware and equipment was thoroughly cleaned for trace organic analysis. Extracts were combined and dried under $N_2$. Samples were re-dissolved in DCM and methylated following the method of Ichihara and Fukubayashi (2010) [66]. Finally, the samples were silylated by adding 20 μL pyridine and 20 μL BSTFA and placing in an oven at 60 °C for 20 min; 50 μL of ethyl acetate was added and the samples analysed using gas chromatography mass spectrometry (GC/MS). Total lipid extracts were analysed on a Shimadzu QP 2010 Ultra gas chromatography mass spectrometer (GC/MS). Separation was performed on a Zebron ZB-5HT column (30 m × 0.25 mm × 0.10 μm) with the helium carrier gas flow at 1.5 mL per minute. Samples were injected splitless onto the column at 40 °C, ramping to 180 °C at a rate of 15 °C/min,

followed by ramping to 325 °C at a rate of 4 °C/min with a final hold for 15 min. The run time totalled 60 min. The MS was set to scan from 50 to 800 m/z with an event time of 0.7 s and a scan speed of 1111 u/s. Concurrently a SIM scan was set to analyse, with an event time of 0.2 s, the transitions *m/z* 191, 205, 217, 412, 414 and 426, indicative of sterane and hopanoid lipid fragments. All peaks were background subtracted and identification confirmed using a combination of the National Institute of Standards and Technology (NIST) GC/MS library database and spectra in literature. Hopanoids and steranes were identified based on the mass spectra from Gallegos (1971) [67]. Identification of alkanes was based on a C21-C40 alkane standard (Sigma-Aldrich, Saint Louis, MO, USA). Contamination was not introduced into the samples during work up as blank samples, worked up concurrently with the fractions, had results comparable to the ethyl acetate instrument blank. The lipid extracts were subsequently analysed for lipid specific $\delta^{13}C$ by gas chromatography–isotope ratio mass spectrometry (GC-IRMS) using a Thermo Delta V Plus mass spectrometer coupled to a Trace Ultra GC, GC Isolink II and Conflo IV. Samples were injected using a Programmable Temperature Vapouriser (PTV) into helium carrier gas. The GC programme had an initial temperature of 100 °C and was ramped for 20 °C/min up to 250 °C and subsequently for 5 °C/min up to 340 °C with a hold time of 18 min.

## 3. Results

### 3.1. Whole-Rock Geochemistry

Whole-rock geochemical analyses of major, minor and trace elements are presented in Table 1. Cape Vani MnMISS samples contain from 6.41 to 29.58 wt.% Mn and 0.61 to 7.98 wt.% Fe. Other important cations include K (2.47–6.41 wt.%), Ca (0.08–0.43 wt.%), Mg (0.06–0.29 wt.%). The analysed samples contain low concentrations of Zn (589–6884 ppm), Pb (90.7–>10,000 ppm), Cu (12.17–2423 ppm), As (211–3551 ppm), Sb (8.1–415.9 ppm), Sr (449–2232 ppm), Co (2.5–78.6 ppm), Ag (9.5–43.6 ppm), Cd (0.5–40.9 ppm), W (4.2–157.2 ppm), and Mo (10.8–188.8 ppm). Finally, they contain low concentrations of Rb (67.7–127.4 ppm), Li (6.0–52.0 ppm), Zr (11.3–56.7 ppm), V (1.0–91.0 ppm), Cr (13–30 ppm), Ni (3.4–13.6 ppm), Ce (8.0–47.0 ppm) and Y (4.2–59.7 ppm).

A number of geochemical classifications have been created that discriminate among marine hydrogenetic, Fe-Mn crust/nodule, diagenetic and hydrothermal origins for Mn oxides [68,69]. Accordingly, the MnMISS samples cluster within the diagenetic and hydrothermal marine fields based on the Mn-Fe-(Ni + Cu + Co)*10 ternary diagram (Figure 2). Fe/Mn ratios for MnMISS samples show high fractionation of Fe and Mn averaging 0.17 (range: 0.07–0.56) typical of hydrothermal Mn oxides from active volcanic arcs, supporting a hydrothermal metal source [70,71]. Concentrations of $SiO_2$ (18.41–66.71 wt.%), $Al_2O_3$ (2.07–5.32 wt.%), and Ti (0.038–0.201 wt.%) may indicate detrital and authigenic silicates, and Ba (4328–>10,000 ppm) barite [69].

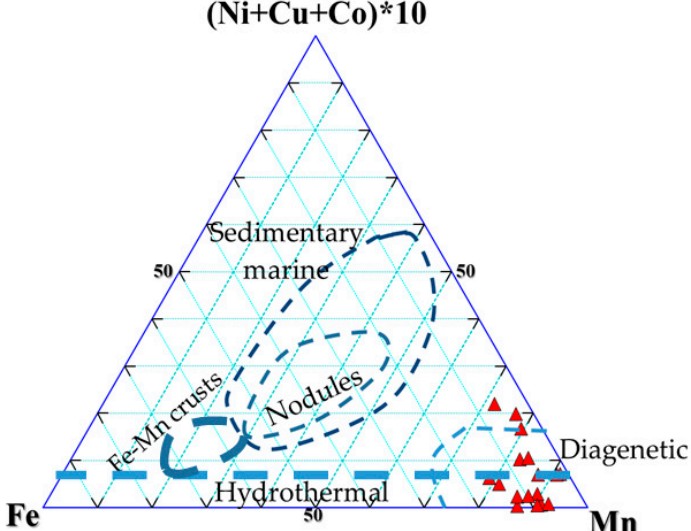

**Figure 2.** Mn-Fe-(Ni + Cu + Co)*10 ternary geochemical classification diagram for discrimination of Mn oxides of marine hydrogenetic, Fe-Mn crust/nodule, diagenetic and hydrothermal origin. Fields after Bonatti et al. (1972) [68], Conly et al. (2011) [69], and Polgari et al. (2012) [37]. Red triangles denote MnMISS data points. Results for Cape Vani Mn oxides indicate a mixed hydrothermal and diagenetic affinity. Note that very few samples fall outside any of the outlined fields, indicating a trend to more diverse origins (see text for discussion).

### 3.2. The Mn Oxide Cement: Microtextures and Mineralogy

MnMISS Mn enrichment is confined to Mn oxides, which typically construct texturally diverse cements to the volcaniclastic detritus; the latter, based on powder X-ray diffractometry (PXRD), transmitted light petrography and SEM-EDS analysis, consists chiefly of K-feldspar and sericitized plagioclase, chloritized biotite, pyroxene and vitric clasts, minor dacitic/andesitic clasts, and, glauconite and barite (Figure 3).

### 3.2.1. Microscopic Textural Diversity

Textural diversity of Mn oxide cement comprises laminated, botryoidal/colloform, and homogeneous dense layer textures. On a microscopic scale, the degree of Mn mineralization ranges from minor patchy Mn cement, to extensive cement-supported textures (Figure 3A,B) and further to heavy mineralization where Mn minerals have cemented almost all detrital grains. Mn oxide cemented granules with organismal features resembling pennate diatoms or benthic algae up to 250 μm long and ~20 μm wide, and possible sponge spicules also occur (Figure 3C). Cementation of mineral grains (e.g., K-feldspar) seems sporadic compared to that of the diatoms/algae-like features.

Mat texture-forming lamination, similar to that observed in MISS [64,72] includes: (1) differentially porous and fibrous Mn oxide microlaminae alternating with dense finely crystalline or cryptocrystalline Mn oxide microlayers, colloform/botryoidal layers (Figure 3B) or planar layers of massive crystalline Mn oxides consisting of pyrolusite (Figure 3D); (2) Mn oxide matrix-supported laminae up to 1 mm thick that are spotted by tufted microstructures, which reach 500 μm to 1500 μm in height (Figure 3E); (3) Pseudocolumnar structures and (micro)stromatolitic fabrics of Mn oxides (Figure 3F), reminiscent of deep sea Mn nodules [73]; and, (4) 100–1000 μm thick bedding plane-parallel and laterally continuous Mn oxide matrix-supported laminae (Figure 3G), which appear either bent, exhibiting smoothly curved upper boundary and uneven and diffuse lower edges (Figure 3G), or as coherent laminae (Figure 3H,I).

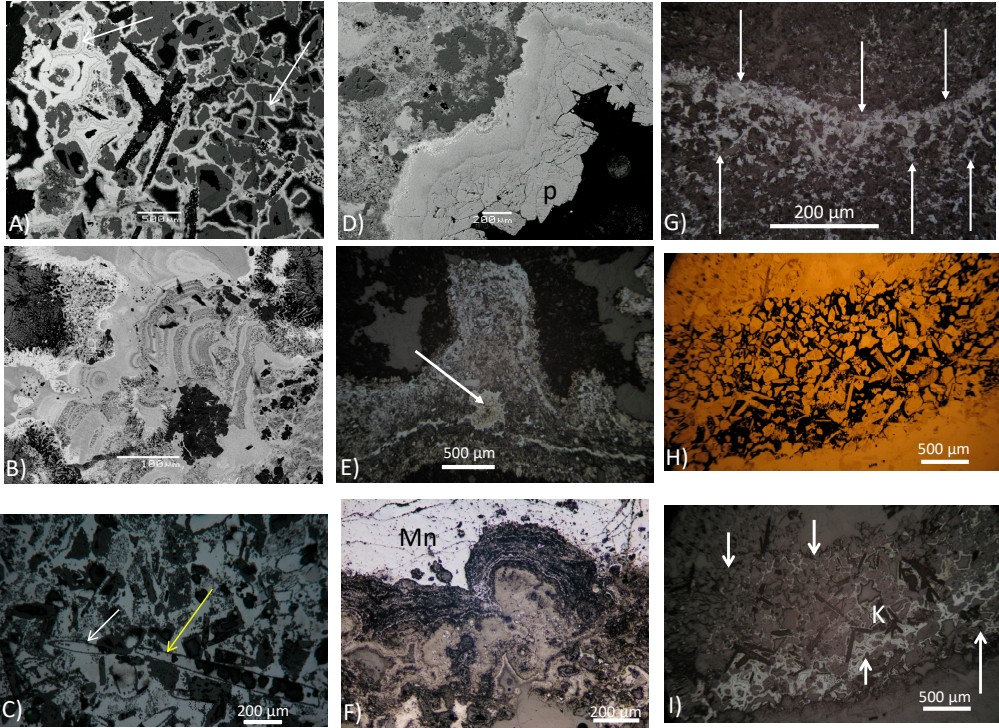

**Figure 3.** Scanning electron microscope (SEM) backscatter electron images (**A,B,D,E,F**) and reflected light photomicrographs(**C,G,H,I**) of Mn-oxide microtextures (**A**) Highly mineralized MnMISS layer with Mn oxide cement-supported texture (left side) grading into grain supported mineralized layer having minor Mn-oxide cement (right side). White arrows for Mn cement. (**B**) Colloform layering of Mn oxide cement. (**C**) Mn oxide cemented MnMISS, reflected light; possible diatoms/algae (white arrows) and possible sponge spicules (yellow arrow) are partially to completely coated by microcrystalline Mn-oxides, the other detritus are K-feldspar and/or unaltered glass fragments with Mn coatings. (**D**) Very fine wavy layers of microcrystalline Mn-oxide lining walls of void space, overgrown by planar layer of massive crystalline pyrolusite (*p*). (**E**) Single tufted microstructures characterized by asymmetric orthogonal to domal thickening of Mn oxide-matrix supported lamina. Note that these microstructures are linked by thin continuous non-isopachous Mn laminae. Note microcrystalline Mn oxide-filled void in the central part (white arrow plus encircled). (**F**) Pseudocolumnar microstromatolite-like structures with microcrystalline (pale) and amorphous cryptocrystalline (dark) Mn oxide laminae, which overgrow bulbous Mn oxides (**B**); K: framework grains. Microstromatolites are overlain by anhedral microrystalline Mn oxide (Mn). (**G,H,I**) Photomicrograph in reflected light (**G** and **I**) and transmitted light (**H**) of bedding parallel Mn-oxide cement supported laminated texture; the white arrows point to the margins of the lamina.

### 3.2.2. Mineralogical and Textural Characterization of Todorokite

Powder XRD (PXRD) data (Table S1) confirm previous mineralogical investigations [41–43,46,48]. The presence of micro- and nano-crystalline todorokite cement has been revealed for the first time by means of PXRD, BSE-EDS, FEG-SEM and HR-TEM investigations. Todorokite is a common tunnel-structured Mn oxide phase present in diagenetic Mn nodules and marine hydrothermal Mn deposits [28]. Todorokite is thought to result from sediment diagenesis and transformation of initial biotic or abiotic poorly crystalline, highly disordered, and layered phases that are structurally similar to δ-MnO$_2$ (vernadite) or birnessite. Interestingly, the key driving factor for this layer-to-tunnel phase transformation of Mn oxides is redox reactions through electron transfers between microbes, minerals, organics, and metals; these reactions occur at natural oxic–anoxic interfaces, for instance those in marine sediments [5,74–78]. The XRD analyses display characteristic diffraction peaks at ~9.6 Å, ~4.8 Å, ~3.2 Å, ~2.45 Å, ~2.2 Å, ~1.9 Å, ~1.7 Å and ~1.5 Å indicative of todorokite (Figure S2) [71,75,76,79]. In

addition, manjiroite was detected by X-ray diffraction reflections at ~7.0 Å, ~3.1 Å, and 2.4 Å (Figure S3) [80]. SEM-backscatter electron (BSE) imaging reveals morphological domination by elongated fibers (>5.0 μm) of crystalline todorokite (Figure S4A), and FEG-SEM imaging shows nanocrystalline todorokite in the form of fibrous needles (<0.4 μm) (Figure S4B). SEM-energy-dispersive X-ray spectroscopy (EDX) analyses of the fibres (Table S2) are consistent with todorokite [80,81], composed of 76–82 wt.% Mn(III/IV)$O_2$. FEG-SEM EDS elemental maps of todorokite nanoplatelets are shown in Figure S5.

The high-resolution transmission electron microscopy (HRTEM) micrographs (Figure 4) provide additional identification of typical todorokite nanocrystalline morphology [75,82–84]: (i) striated nanoplatelets (~10–20 nm wide), either isolated or aggregated together laterally elongated longitudinally up to 200 nm in length (Figure 4A,B); (ii) structural defects in the form of spiral-like longitudinal dislocations (Figure 4C,D); (iii) trilling intergrowths (Figure 4A,B); (iv) structural defects in the form of tunnel width inconsistencies (Figure 4E).

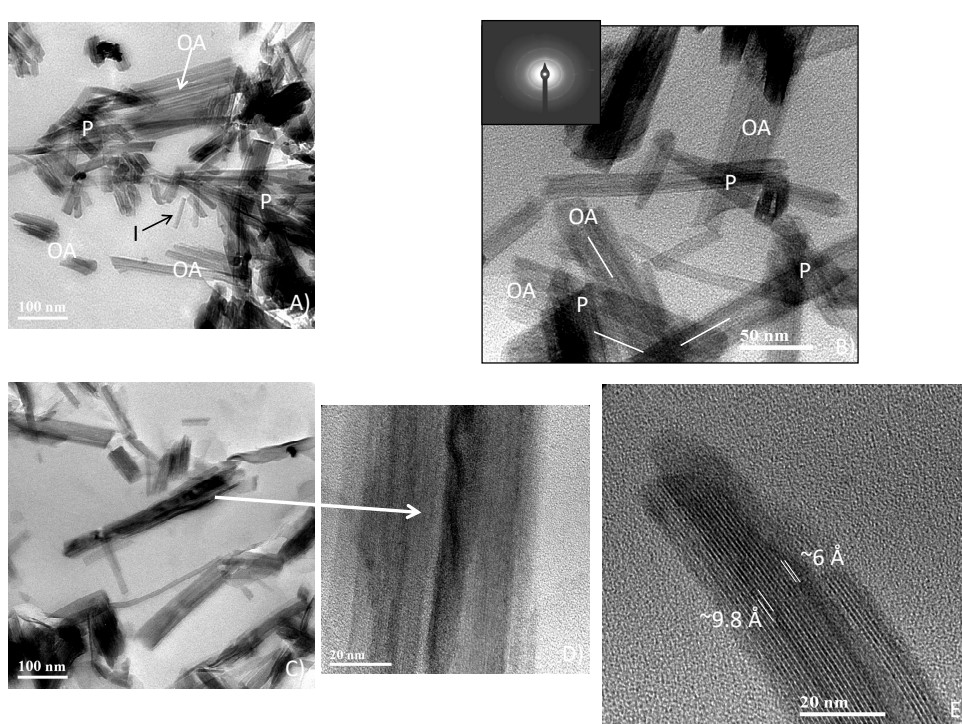

**Figure 4.** High–resolution transmission electron micrographs (HRTEM) of nanocrystalline todorokite: (**A,B**) (I) Isolated primary elongated striated particles of acicular todorokite; (OA): acicular particles elongated up to 200 nm, which are laterally aggregating together, forming stable laths up to 100 nm wide; (*p*): plate-like morphology, with plates comprised of overlapping todorokite laths—note trilling intergrowths where individual laths are oriented specifically in three distinct directions at 60°/120° (white lines in Figure 4B) to each other to form large aggregates with a plate-like morphology. The selected area electron diffraction (SAED) pattern in b shows diffraction rings at 2.4 Å. (**C,D**) laterally aggregated todorokite laths comprised of aligned overlapping elongated nanoparticles, also showing in the enlarged area structural defects in the form of spiral-like longitudinal dislocations; (**E**) todorokite showing lattice fringes at a spacing of approximately 9.8 Å; note tunnel-width inconsistencies (heterogeneous tunnel dimensions) observed as additional ~6 Å spacings in the a direction.

## 3.3. Electron Paramagnetic Resonance (EPR) Study

EPR measurements on Mn oxides (Figure 5) were performed on samples from the Mn mineralized barite ± silica – rich feeder veins (Figure S1A—"Vani vein" in Figure 5), MnMISS, i.e., Mn nodular crusts (Figure S1B—"Vani nodules" in Figure 5), Mn mats (Figure S1C—"Vani crust" in Figure 5),

and the chimney structure outer surface (Figure S1F—"Vani vent" in Figure 5). The latter three represent structures developed under shallow-seafloor conditions [46], while the first represents deeper settings in the hydrothermal system. EPR measurements of the Mn oxides from the feeder veins gave a signal centered at g = 2.29, and a line width (peak-to-peak) of approximately Δpp = 1900 G (Figure 5). Measurements of the Mn nodular crust gave a signal centered at g = 2.01, and a line width (peak-to-peak) of approximately Δpp = 1200 G. The Mn-nodules gave a signal centered at g = 2.05, and a line width (peak-to-peak) of approximately Δpp = 1300 G. Finally, the chimneys gave a signal cantered at g = 2.09, and an approximated line width of 1100 G.

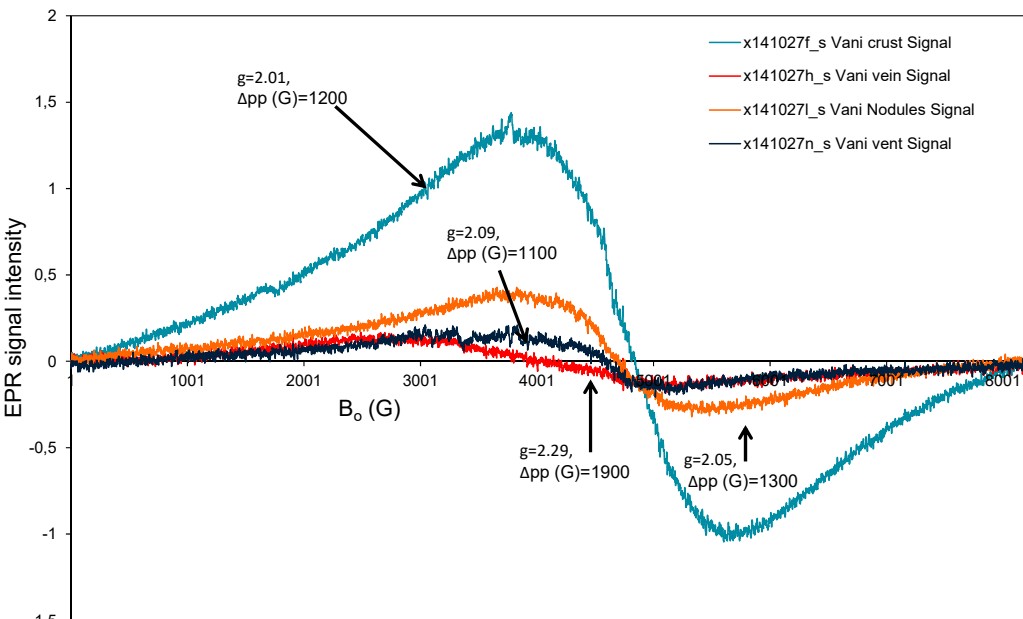

**Figure 5.** Electron paramagnetic resonance (EPR) spectra for Cape Vani Mn oxide samples, i.e., crusts and Mn mats (Vani crust), chimneys (Vani vent), Mn-nodules (Vani Nodules), and feeder veins (Vani vein). Also shown are g values and Δpp (G) for each sample type. (G is the unit gauss; g is the (electronic) g factor (dimensionless)).

### 3.4. Total Organic Carbon [$TOC_{org}$ (%)] and Lipid Biomarker Analysis

Total organic carbon contents ($TOC_{org}$ as wt.%) range from 0.011 to 0.046 wt.% (Table 2). The most abundant lipid types in the lipid extracts were fatty acids methyl esters (FAME) (Table 2; Figure 6A). Saturated FAMEs in the range of C14:0 to C19:0 were the most abundant, although a methyl branched C15:0 dominated the chromatogram in some samples. Mono and polyunsaturated FAMEs and a long chain C24:0 fatty acid were also identified, but low in abundance. *Iso* and *anti iso* branched chain fatty acids, biosynthesised by bacteria using valine or leucine as the precursor molecule of fatty acids [85], were found in five samples but low concentrations prevented obtaining their specific $\delta^{13}C_{org}$ signals. Alkanes, Trimethylsilyl (TMS) derivatives of triterpenoids and hopanoids comprised <20% of the total lipid fraction. Two samples contained only fatty acids and some gave no discernible signal (data not shown), suggesting a high degree of degradation of organic molecules at certain sites. The hopanoid range was not the same in all samples, further indicating some difference in preservation between sites. Sterols, (cholesterol and campesterol), biosynthesised by both plants and algae, were identified in two samples where the most abundant triterpenoid was cholesterol. For comparison Figure 6B shows a modern day active hydrothermal sediment from Spathi Bay, S. Milos [86]. The same major lipid groups and types are seen in Figure 6A,B but an increased number of low molecular weight lipids and triterpenoid lipids, presumably the less recalcitrant fraction, are observed in the modern day sediments.

**Table 2.** Summary of bulk $\delta^{13}C_{org}$ (‰), $C_{org}$ (%) and lipid biomarker analysis. Lipid specific $\delta^{13}C_{org}$ (‰) values are shown in parenthesis.

| Sample No (MISS Type) [1] | Bulk $\delta^{13}C_{org}$ vs. PDB (‰) | $C_{org}$ (%) | FAME range | *iso/anti iso* FAME | Alkanes | Hopanoids | Terpenoid Alcohol/Sterols/Sterane | Lipid Origin |
|---|---|---|---|---|---|---|---|---|
| MI-04–27 (roll-up structure) | | | | *i*C12:0, *i*C15:0, *ai*C15:0 | | | | |
| MI-10–15 (Roll-up structure) | −22.72 | 0.022 | | | C35-C37 | C29-C35 | Amyrin | Prokaryote/Eukaryote |
| MI-10–22 (mat fragments and chips) | −23.57 | 0.015 | C16:0-C24:0 (−32.0 to −28.0 ‰) | | | C29-C34 | | Prokaryote/Eukaryote |
| MI-10–24 (fossil gas dome) | −22.91 | 0.015 | C16:0-C18:0 | | | C29-C35 | | Prokaryote |
| MI-10–27 (upturned/curled margins) | −21.51 | 0.017 | C16:0-C18:0 | | | C29-C32 | Cholesterol TMS, Neoergosterol, Campesterol, β-cholesterol TMS | Prokaryote/Eukaryote |
| MI-10–27a (upturned/curled margins) | −22.0 | 0.021 | C14:0-C18:0 (−29.9 to −29.0 ‰) | | | C29-C34 (−47.2 to −51.00 ‰) | | Prokaryote |
| MI-10–29 (Mn nodules) | −24.9 | 0.014 | C16:0-C18:0 | | | C29-C35 | | Prokaryote |
| MI-10–31 (wrinkle structures) | −25.53 | 0.011 | C16:0-C18:0 | *i*C17:0 | | C29-C35 | | Prokaryote |
| MI-10–33 (wrinkle structures) | −25.0 | 0.015 | C15:0-C19:0 | *i*C19:0 | | C29-C35 | | Prokaryote |
| MI-10–34 (wrinkle structures) | −25.4 | 0.017 | C16:0-C18:0 (−30.2 to −30.7‰) | | | C29-C35 (−39.2 to −37.4‰) | 5-α-Cholestane | Prokaryote/Eukaryote |

**Table 2.** *Cont.*

| Sample No (MISS Type) [1] | Bulk $\delta^{13}C_{org}$ vs. PDB (‰) | $C_{org}$ (%) | FAME range | *iso/anti iso* FAME | Alkanes | Hopanoids | Terpenoid Alcohol/Sterols/Sterane | Lipid Origin |
|---|---|---|---|---|---|---|---|---|
| SMO-05–10 (mat fragments and chips) | −23.43 | 0.018 | C16:0-C18:0 | | | C29-C31 | | Prokaryote |
| VA-05–10 (growth bedding) | −27.34 | 0.011 | C16:0-C18:0 (−27.1 to −26.3‰) | *i*C15:0, *ai*C19:0 | C29-C31 (−29.0 to −24.3‰) | C29-C31 (−48.2 to −38.6‰) | | Prokaryote/Eukaryote |
| VA-05–16 (Mn nodules) | −25.57 | 0.012 | C16:0-C18:0 | | | C30, Diploptene | Cholest-5-en3β-ol, 23-methyl cholesta-5,22-dien3β-ol, 24-methyl cholesta-7,22-dien3β-ol, 24-methyl cholesta-en3β-ol, 24-ethyl cholesta-5,22E-dien3β-ol | Prokaryote/Eukaryote |
| VA-05–18 (growth bedding) | −27.19 | 0.011 | C16:0-C18:0 | | | | | Unknown |
| VA-05–20 (growth bedding) | −27–19 | 0.011 | C12:0-C18:0 (−31.9 to 29.4‰) | | | | | Eukaryote |
| SMO-05–09 (growth bedding) | −26.38 | 0.046 | C14:0-C18:0 | *i*C15:0 | | | | Prokaryote |

[1] MISS type according to Kilias (2012) [46].

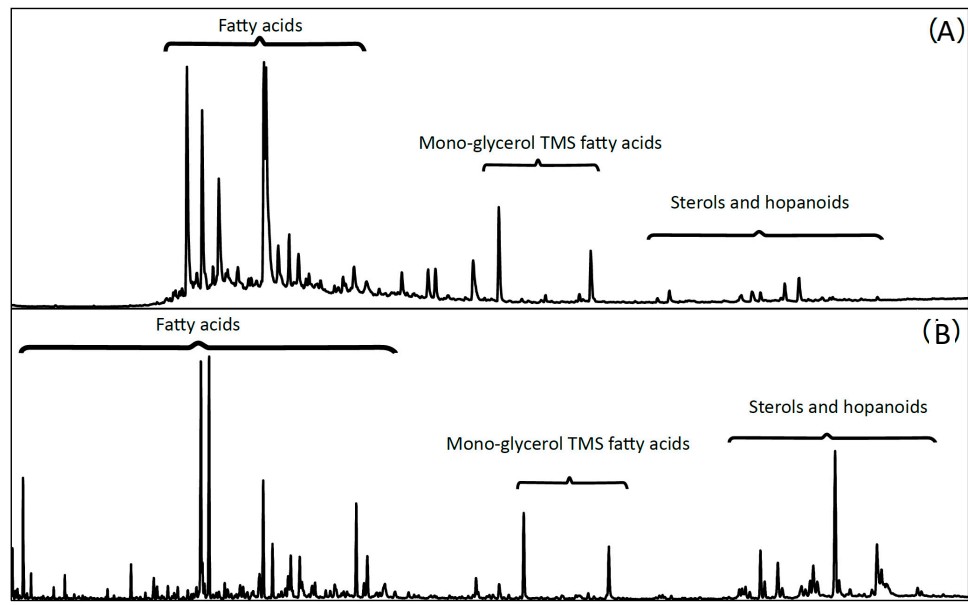

**Figure 6.** Gas chromatography mass spectrometry (GC/MS) chromatograms of lipid extracts showing major lipid classes identified in (**A**) Quaternary MnMISS samples. For comparison, (**B**) a modern active hydrothermal sediment from Spathi Bay, S. Milos [86].

### 3.5. Carbon Isotopes [Bulk $\delta^{13}C_{org}$ (‰), and Lipid Specific $\delta^{13}Corg$ (‰)]

Bulk $\delta^{13}C$ data and lipid specific $\delta^{13}C$ data associated with the various Mn ore types are given in Table 1. Bulk $\delta^{13}C$ varies from −27.19‰ to −21.51‰. Lipid specific data for the FAMEs, alkanes and hopanoids ranged from −28.0‰ to −32.0‰, −24.3‰ to −29.0‰ and −37.4‰ to −51.0‰, respectively.

## 4. Discussion

The emphasis of this paper is to decipher the link between Mn oxide formation, low-T hydrothermal fluid flow, and microbially induced sedimentary structures (MISS) from Early Quaternary shallow-marine/tidal-flat siliciclastic sandstones/sandy tuffs of the CVSB. To the best of our knowledge, Mn oxide mineralization preserved in the siliciclastic sedimentary rock record has never been spatiotemporally and genetically linked to the growth and preservation of MISS, especially not under sunlit low-temperature hydrothermal metal exhalation conditions [2,3,5–7,29].

The previous approach for recognizing Cape Vani MISS [46] will be reassessed on the basis of newly established biogenicity criteria for identification of "proven MISS", coined by Davies et al. (2016) [87] (see Section 4.1.). Subsequently, we will discuss Mn oxide deposition as the result of: (1) direct activity of microorganisms, i.e., microbial oxidation of Mn (II) during MISS growth leading to "biomineralization"; (2) synsedimentary/pre-burial diagenetic processes associated with MISS preservation. These refer either to Mn oxides being a by-product of organic matter produced by microbial degradation interacting with inorganic compounds (e.g., hydrothermal Mn) to form Mn oxide mineral precipitates during "organomineralization" [88], or replacement of decaying microbial organic matter by minerals as a component of "destructive biomineralization" [64] or "mat decay mineralization" [89] (see Section 4.2).

### 4.1. Biogenicity of the Cape Vani Microbially Induced Sedimentary Structures (MISS) Revisited

The investigated Mn-mineralized sedimentary structures have been earlier characterized as MISS [46], based on biogenicity criteria reviewed in Noffke (2010) [64]. More recently, Davies et al. (2016) [87] noted that MISS should be treated with caution as they are a subset of sedimentary surface textures (SST) that comprise those of abiotic origin. They proposed a new classification scheme that categorises the degree of certainty in assigning SST to MISS, based on the weight of accessory evidence:

i.e., Category B signifying definitively biogenic/microbial (proven MISS), and Category A definitively abiotic (proven abiotic); Category ab where there is uncertainty, and Category Ab or Category Ba where there is uncertainty but one interpretation is supported. Accordingly, the previous approach for recognizing MISS at Cape Vani [46] is reassessed below on the basis of all the biogenicity criteria coined by Davies et al. (2016) [87], which include previously missing ones: (1) preserved organic material, i.e., lipids; and, (2) geochemical biosignatures, i.e., bulk and lipid specific $C_{org}$ isotope biomarkers. As detailed below, based on existing field [46] and new micro-textural and geochemical criteria (this study), the Cape Vani Mn-mineralized SST fulfil all the criteria for biogenicity in Davies et al. (2016) [87], and are therefore defined as Category B MISS.

### 4.1.1. Similarity of Form to Modern MISS

The Cape Vani MnMISS bear striking similarities to structures produced by modern microbial mats in their host sandy sediments [90,91], which include:

1.　Sedimentary features and structures related to growth, such as growth bedding (see Figure 5A in Kilias (2012) [46]), and nodular to biscuit–like surface structures (see Figure 5C in Kilias (2012) [46]).
2.　Structures generated from a change in growth direction, such as tufted microstructures (Figure 3E), macroscopic reticulated surface patterns resembling 'elephant skin' (see Figure 6 in Kilias (2012) [46]), and microscopic pseudocolumnar structures and microstromatolite-like fabrics of Mn oxides (Figure 3F). The tufted microstructures (Figure 3E) compare strikingly well in size and shape with tufted biofilm structures left by modern cyanobacterial mats in their host sediments [90–93], as well as experimental tufted cyanobacteria biofilms [94]. The tufted morphology suggests coordinated growth commonly known from cyanobacterial mats in modern environments [93]. Consequently, tufted microstructures may be interpreted as a result of vertical growth of phototactic cyanobacteria stabilized by EPS. This is supported by the Mn oxide-filled bubble-like structure (white arrow in Figure 3E) similar to oxygen-rich bubbles trapped within modern oxygenic cyanobacterial mat fabrics [93].
3.　Features and fabrics related to trapping and binding of sediment, such as microbial lamina-bound volcaniclastic detrital grains (Figure 3H,I).
4.　Features derived from physical mat destruction, such as desiccation and cracks with upturned and curled margins, and jelly rolls and cracks (see Figure 7 in Kilias (2012) [46]), gas bubbles aerating the jelly mat below supporting mat detachment (see Figure 9D in Kilias (2012) [46]), and erosional edges, chip production (see Figure 10 in Kilias (2012) [46]).
5.　Features derived from mat decay and diagenesis, such as punctured gas domes, indicating sealing by mats interacting with diagenetic gas production (see Figure 9A,B in Kilias (2012) [46]).

### 4.1.2. Distribution Reflecting the Hydrodynamic Conditions of the Depositional Environment

Distinct sedimentary structures detected in the CVSB [46], combined with fluid inclusion evidence [51], indicate that the identified MnMISS have been deposited in a shallow-marine (<50 m) tidally-influenced environment. Such depositional environments experience a complex range of hydrodynamic conditions, and sediment biostabilization by microbial mat effects [64]. The patchy distribution of the identified MnMISS, combined with their large morphological variety [46] suggest reflection of the local hydrodynamic conditions of the depositional environment. This is supported by modern MISS distribution models in siliciclastic tidal environments [93,95]: e.g., mat fragments and chips (see Figure 10 in Kilias (2012) [46]) found in the lower intertidal zone, cracks with upturned and curled margins (see Figure 7 in Kilias (2012) [46]) in the upper intertidal to lower supratidal zones, gas domes (see Figure 9 in Kilias (2012) [46]), and tufted microbial mats (Figure 3E) in the upper intertidal to supratidal zones.

4.1.3. Preserved Organic (Lipid) Material and Bulk and Lipid-Specific $C_{org}$ Isotope Biomarkers

Geochemical and lipid biomarker analyses show negative bulk and lipid specific $\delta^{13}C$ organic matter ($C_{org}$) values, and a prevalence of preserved prokaryotic organic biomarkers, which suggest that the analyzed MnMISS formed in an environment predominated by microbial activity (Table 2). Light $C_{org}$ isotope compositions have been used to support a microbial interpretation of putative MISS [96–98]. The MnMISS display bulk and lipid-specific $\delta^{13}Corg$ signatures, which are comparable to those extracted from the modern day Mn oxide (birnessite) depositing non-cyanobacterial mats (−18‰ to −20‰, and −19.7‰ to −22.6‰, respectively), in shallow-seafloor hydrothermal sediments from Spathi Bay, SSE Milos coast [86]. In addition, the modern mats, are composed entirely of microbial guilds that fix and utilise carbon primarily by the Calvin cycle [86]. Consequently, similarities in $\delta^{13}C_{org}$ between the modern and past environments suggest primary carbon fixation mainly associated with the Calvin cycle in the MnMISS. Furthermore, the modern brown-capped sediments (Mn and Fe oxide-rich) share similar biomarkers with the MnMISS (Figure 6). Our lipid data strongly suggests that the system in which the MnMISS structures were created is similar to the processes that are currently occurring in the Spathi bay Mn-rich hydrothermal sediments. In addition, data (which will be made available in an independent communication) acquired by genetic fingerprinting of the Mn oxide-rich zones of the modern mats, using DNA specific markers, suggests that the modern Mn-precipitating mats are associated with Mn (II)-oxidizing *Bacillus* species related to lineages found in the Guaymas hydrothermal vent field, California [99], and widespread $\alpha$-Proterobacterial Mn(II)-oxidizing genes [100].

Where obtainable, lipid-specific hopanoid $\delta^{13}C$ values ranged from −37.4‰ to −51.0‰, the more depleted values of which indicate an origin from methanotrophic bacteria oxidizing methane aerobically [101,102]. Experimental studies have demonstrated that a similar range of hopanoid $\delta^{13}C$ values are biosynthesised by methanotrophs [103,104], and similar fatty acid $\delta^{13}C$ values have been found to originate from aerobic methanotrophs inhabiting shallow marine microbial mats [105]. The depleted $\delta^{13}C$ values indicate cyanobacteria are an unlikely source for the hopanoids. This view is further supported by the absence of alkanes <*n*-C29:0, alkenones, heptadecane (*n*-C17:0) and hop-22(29)-ene, which often dominate lipid extracts from phototrophic microbial systems [102,106–109]. The absence of those specific biomarkers may suggest that photorophic cyanobacteria are not a key component of the sampled MnMISS; however, biomarkers may have been actively recycled (possible in a low-carbon environment, Table 2) and re-metabolised by microorganisms.

The occurrence of long chain alkanes with specific $\delta^{13}C$ values in the range of −24.3‰ to −29.0‰, together with sterols, strongly suggest an origin from C3 plants. Plant lipids described by Callac et al. (2017) [86] in the modern day Spathi Bay hydrothermal sediments (Figure 6B), where seagrasses enriched their membrane lipids in $^{13}C$ by up to a factor of two, relative to the bulk sediment. Also comparable were the FAME values −31.6‰ to −29.5‰ reported in brown-capped sediments, in the modern day Spathi bay sediments. FAMEs in the MnMISS samples ranged from −32.0‰ to −26.3‰ further suggesting similar organisms and metabolic processes were present in both the modern and past systems. The lack of cyanobacterial biomarkers shows that oxygenic photosynthesis is not an important process in carbon fixation in this environment. Therefore, the biomarker lipid analyses of the Cape Vani MnMISS, suggest a prokaryote-dominated environment, including some aerobic methanotrophy, with influence from preserved marine plant lipids. A vast range of bacteria, including the $\alpha$-Proteobacteria to which most Mn oxidizing bacteria and type II methanotrophs belong, produce the identified hopanoids [5,100]. Thus, on the basis of the biomarker lipids and their carbon isotopic values, we can infer a strong bacterial association of MnMISS with methanotrophy. If photoautotrophic cyanobacteria participated in the mat-constructing microbial group of the studied MnMISS, then these may have produced $CH_4$ and sustained methanotrophy in the CVSB sunlit and oxic environment [110]. In any case, even if present, cyanobacteria are not a key component of the sampled MnMISS.

In summary, our data suggest that MISS are not unique to cyanobacteria mats, and methanotrophs may have contributed significantly to the formation of the MnMISS. This view is further supported by

the ability of methanotrophs to form mats in seep systems [105,111,112]. As demonstrated, the extracted biomarker content of the ancient Mn mats strongly resemble those of the non-cyanobacterial Mn-depositing mats on the modern shallow-seafloor hydrothermal setting on the Milos coast (Figure 6). These mats are predominated by genes associated with the Calvin cycle and display bulk and compound-specific $\delta^{13}$Corg signatures comparable to those extracted from the uplifted ancient MnMISS [86]. Similarly, the methanotrophs prominent in the modern mats [86] are experimentally known to produce hopanoids containing $\delta^{13}$C values in the range recorded in the present study [103,104]. Furthermore, the data suggest that unlike the patchy appearance of the contemporary mats, resulting from weak and diffuse hydrothermal activity, similarly widespread mats on the Early Quaternary seafloor precipitated Mn released into seawater as a result of intense volcanic hydrothermal activity.

### 4.1.4. Textural Evidence of Sediment (Bio) Stabilization

Macroscopic textures indicating sediment (bio) stabilization [64] have been identified in the CVSB, and include: roll-ups, shrinkage cracks, gas domes and mat chips (see Figures 7–10 in Kilias (2012) [46]). In addition, features related to siliciclastic sediment biostabilization were revealed by microscopic examination of MnMISS. These are expressed as laminated fabric containing alternation of laminae that are enriched in Mn oxide minerals with laminae that consist largely of siliciclastic grains and trace Mn oxides (Figure 3). Texturally, Mn oxide-rich laminae consist of oriented siliciclastic grains (mainly quartz) "floating" within a botryoidal/colloform Mn oxide cement/matrix (Figure 3H,I). These microscopic textures compare strikingly well to microbial biolaminated structures in modern microbial mats, which are related to binding and stabilization of sediment (see Figures 2-1-7 E in Gerdes, 2007 [91]). The modern structures are characterized by microbial mat matrix-supported quartz grains, which are enveloped and rotated to an energetically suitable position by the growing microbial mat resulting in oriented grains with their long axes parallel to the lamination [64,91,113]. In the Cape Vani MnMISS, oriented framework grains (i.e., quartz, K-feldspar) are preserved in a fine patchy botryoidal/colloform Mn oxide cement, in place of the microbial filaments cluster soft matrix surrounding the grains of the modern textures (Figure 3H,I).

The positive identification of microtextural evidence of sediment (bio) stabilization as illustrated in Figure 3H,I, meets the final criterion for biogenicity of Mn-mineralized MISS (MnMISS) in the sedimentary deposits of the CVSB.

The Cape Vani Mn mineralized sedimentary surface textures fulfil all these criteria, and are therefore proved to be Category B MISS, i.e., demonstrably biotic (microbial) [87].

### 4.2. Biogenicity and Syngenicity of Mn–Mineralized Microbially Induced Sedimentary Structures (MnMISS)

#### 4.2.1. Biogenicity

A suitable means of assessing microbial-mediated oxidation of Mn (II) and subsequent biomineralization of Mn oxide minerals in the MnMISS, is using EPR spectroscopy [114,115]. Kim et al. (2011) [114] studied a wide range of Mn oxides including synthetic Mn oxides, natural Mn oxides with both a biological and abiotic origin, and bacteriogenic Mn oxides from controlled laboratory experiments and showed that biogenic Mn oxides has EPR spectral signatures distinct from abiotic Mn oxides. They also showed that natural Mn oxides with a suspected biological origin had EPR signatures distinct from pure abiotic Mn oxides, and thus suggested that EPR could be used to distinguish between biological and abiotic Mn oxides in natural samples. Ivarsson et al. (2015) [115] further showed that Mn oxides associated with fungal-prokaryotic consortia in subseafloor basalts had a biogenic EPR-signal and were the product of microbial Mn cycling and subsequent biomineralization of Mn oxides.

Depending on their origin, Mn oxides fall within certain categories with respect to the EPR spectral signatures. Abiogenic Mn oxides have linewidths ΔH > 1200 G, suspected biominerals like

desert varnish and Mn nodules have linewidths 600 G < ΔH < 1200 G, and pure biogenic Mn oxides ΔH < 560 G [114]. Also, abiogenic Mn oxides have widely scattered g-values while biogenic Mn oxides cluster around g = 2.0. Thus, the EPR measurements of the Mn mats and the smokers with g at 2.01 and 2.09, respectively, and line widths of 1200 G and 1100 G, respectively, fall within the biomineral category of Mn oxides (Figure 5) [114]. This category represents natural Mn oxides like desert varnish with a strong biological affinity but where the reaction pathway from Mn(II) to Mn (III/IV) oxide mineral is not completely understood. The main difference of this category compared to the category with ΔH < 560 G is that it consists of natural samples and the biological affinity has only been observed in situ, while the latter category has been observed in controlled laboratory experiments. The g value of the nodules of 2.05 (Figure 5) also suggests a biogenic origin even though the G value of 1300 is slightly above the 1200 G boundary for Mn biominerals. The EPR measurements of the feeder veins, on the other hand, with g at 2.29 and a line width of 1900 G (Figure 5) correspond very well to abiotic Mn oxides. Thus, the surficial MnMISS samples have clear EPR signals for biological involvement in the formation of Mn oxides, while the deeper samples from the feeder veins show clear abiotic signals for the Mn oxides. Biological, i.e., metabolic, involvement in the formation of Mn oxides in MnMISS during deposition, or preservation of biogenic EPR signals during pre-burial curing/diagenesis, is thus, indicated by the EPR measurements. Consequently, biogenic Mn oxide precipitation established by EPR spectroscopy combined with hydrothermal fluid flow-induced pre-burial curing in areas of hydrothermal fluid flow and high geothermal gradient, may account for today's crystalline Mn oxide resource.

### 4.2.2. Syngenicity

Except for the issue of biogenicity, crucial to their interpretation of the MISS-hosted Mn oxides is the issue of syngenicity. Did the Mn oxides form during MISS deposition, and/or thermal maturation/pre-burial diagenesis associated with MISS preservation, or were the Mn oxides replaced post-deposition?

A number of observations detailed above support syngenicity. That is they bear remarkable similarities to mineralization features formed by mat-related mineral precipitation, which have been ascribed to mat metabolic pathways in sandstones such as photosynthesis, mat decay and pre-burial diagenesis [16,89,93,116]. Moreover, mat decay-controlled mineralization may be preserved as a thin layer of fine sandstone grains cemented by minerals such as calcite, pyrite, siderite, and Fe-dolomite [89,116]. Syngenesis of Mn oxides is also supported by macroscopic structures such as pre-consolidation slumping of MnMISS (mat-slump structure), and compaction load-cast structures. This interpretation is also supported by associated "roll-up" structures (see Figure 8 in Kilias (2012) [46]). These types of soft sandy sediment deformation structures imply internal cohesive strength suggestive of a microbial mat origin [89]. Furthermore, syngenicity of Mn oxides is indicated by the presence of nano-crystalline and crystal-defective todorokite (Figure 4; Figures S4B and S5). Todorokite with these textural features has been quoted as a product of bacterial, and/or fungal, Mn (II) oxidation [5,115], or a product of early diagenetic transformation of "relatively amorphous" biogenic Mn oxides, or birnessite, under mild hydrothermal conditions [5,74,75]. The overlying seawater column is implicated for the $O_2$ supply, to form todorokite; this is based on the absence of phototrophic microbial lipid extracts and the more depleted hopanoid $\delta^{13}C$ values, which indicate methanotrophic bacteria oxidizing methane aerobically [101,102], and do not support an in situ phototrophic mat source for marine photosynthetic $O_2$ (See Section 4.1.3) The lack of organic carbon (Table 2) indicates this is a low carbon environment. However, hydrothermal systems are dynamic environments with high microbial functional diversity. Therefore, environmental variability may create a sedimentary environment with patchy density of mat and locally low carbon concentrations. It has been shown for modern hydrothermal soft marine sediments subjected to moderate hydrothermal venting that the processing of organic matter is strongly influenced by the production of chemosynthetic organic matter and that this variable depends on the vent site [117].

Finally, Mn oxide cement-supported textures of volcaniclastic and biogenic sandstone from the active Izu–Bonin–Mariana (IBM) arc system, W. Pacific [71], are spectacularly similar to the CVSB shown in Figure 2. The (IBM) textures have been interpreted as "primary stage of Mn oxide mineralization" supporting Mn oxide syngenesis at CVSB.

Individually, the above textural observations do not provide conclusive evidence for either syngenetic mat metabolism, and/or pre-burial curing/diagenetic mat-decay mineralization, which may have influenced Mn oxide deposition and concentration during the formation/preservation of the MnMISS. However, when all the observed textural data above are taken in account, together with ERP evidence, we believe they indicate a clear role for syngenetic (synsedimentary to pre-burial curing/diagenesis) biological processes in incipient Mn sequestration at the sediment–water interface through shallow depth. Mn oxide biomineralization may have proceeded from forming early rim cement on grains to completely filling pore space and finally abiotic intense cementation of grains in the most advanced pre-burial diagenetic stages. Based on our data, an undivided abiotic Mn replacement post-deposition origin [41,42,48] appears less likely.

## 5. Conclusions

Manganese oxide ores of the Cape Vani Mn deposit consisting chiefly of micro- to nano-crystalline todorokite and manjiroite, hollandite, and vernadite, cement Quaternary microbially induced sedimentary structures (MnMISS) in the northwestern most tip of Milos Island, Greece. The MnMISS have been developed along bedding planes of Upper Pliocene to Lower Pleistocene shallow-marine/tidal-flat volcaniclastic sandstones/sandy tuffs closely associated with the Cape Vani paleo-hydrothermal vent field. A biological origin for the MnMISS is supported by their similarities to modern and ancient documented examples of MISS, patchy distribution reflecting local hydrodynamic conditions of the depositional environment, crucial microtextures recognized in thin section and preserved organic (lipid) biomarkers and, bulk and lipid-specific $\delta^{13}$C organic matter ($C_{org}$) biosignatures. Our data suggest that MISS are not unique to cyanobacteria mats, but shallow-marine microbial mats inhabited also by aerobic methanotrophs may have contributed significantly to the formation of the MnMISS. Furthermore, the data suggest that unlike the patchy appearance of the contemporary Mn mats, resulting from weak and diffuse hydrothermal activity, similarly widespread mats on the Early Quaternary seafloor precipitated Mn released into seawater as a result of intense volcanic hydrothermal activity, using the overlying seawater column for the $O_2$ supplyThe surficial MnMISS samples have clear EPR spectroscopy signals for biological involvement in the formation of Mn oxides, while deeper samples from the feeder veins show clear abiotic signals for the Mn oxides.

Our data taken together indicate a syndepositional/pre-burial diagenesis origin of the MISS-hosted Mn oxide mineralization: MISS formation and preservation (i.e., growth, metabolism, destruction, and decay of microbial mats) corresponding to the shallow marine to tidal flat euphotic zone (probably <50 m deep), and hydrothermal fluid-sourced $Mn^{2+}_{aq}$, have cooperatively created the essential biological active environment predominated by microbial activity that has enabled microbially mediated $Mn^{2+}_{aq}$ oxidation and Mn (III/IV) (bio)oxide pore space-filling precipitation. This initial stage has progressed to crystalline Mn oxide-cementing sandstone, and intense cementation of volcaniclastic detritus and early Mn oxides, in the most advanced hydrothermal fluid flow-induced pre-burial curing/diagenesis stages, accounting for today's crystalline Mn oxide resource. The Cape Vani Mn oxide mineralization may serve as a 'biomarker' in the Quaternary siliciclastic rock record, and widen the spectrum of environments and geobiological processes responsible for marine Mn biometallogenesis to include microbial mats inhabited by aerobic methanotrophs.

**Supplementary Materials:** The following are available online at http://www.mdpi.com/2075-163X/10/6/536/s1. Figure S1: Hydrothermal features, and selected Mn ore styles and Mn mineralized microbially induced sedimentary structures (MISS), Cape Vani Mn oxide deposit. (A) Stringer Mn deposits. Mn mineralised barite (±silica) veins stringer networks crosscut stratiform Mn ore, Mn nodules, bedded barite ore, and cross-bedded sandstone/sandy

tuff host (Vani vein/Figure 5). (B) MISS type: Nodules. Loaf- to mound-shaped structures with nodular and cauliflower-like thrombolytic surface fabric (VA-05-16/Table S1). (C) MISS Type: Mat fragments and chips. Plan view of upper bedding surface of unmineralized sandstone/sandy tuff covered by large patches of millimeter-thick Mn crusts, with highly irregular outlines and embayments, crosscut by white smoker–type hydrothermal-exhalative barite (-silica) feeder veinlets (black arrows). Chisel is 21 cm long (SMO-05-10/Table S1). (D) MISS type: Mat-Layer Structure. Upper bedding surface showing "mat layer structure." Concentric rims with crenulated appearance and discoidal colony texture (dotted line) (VA-05-05/Table S1). (E) MISS type: Upturned and Curled Margins. Cracks with upturned and curled margins on upper surface of fine-grained Mn sandstone/sandy tuff. Curved and upturned (involute) margin (arrow) of inferred fossil microbial mat consists of Mn-rich sandstone. Chisel in C is 21 cm (MI-04-27/Table S1). (F) Mn sandstone/sandy tuff-chimney. Tubular cone possibly representing a white smoker structure, consisting of erratic and locally patchy amounts of Mn oxides and barite that cement epiclastic sandstone/sandy tuff; knolls of Mn oxide minerals are adhering to the outer surface and tip of cone (VA-05-06/Table S1). Number of analysed sample/Table with mineralogical composition or Figure with EPR analysis, Figure S2: Representative X-ray powder diffraction pattern obtained for todorokite (T)-rich samples, Figure S3: Representative X-ray powder diffraction patterns obtained for manjiroite(M)-rich samples [80], Figure S4: (A) SEM-backscattered electron (BSE) image of elongated fibres of crystalline todorokite (>5 μm) intergrown within a platy todorokite matrix, which cement K-feldspar microclasts (K); (B) field-emission gun scanning electron microscope (FEG-SEM) image of nanocrystalline todorokite in the form of fibrous needles (<0.4 μm) intergrown within a platy Mn oxide matrix; these fibres appear to be aggregated into a dense network of fibres displaying a plate-like growth morphology, which is characteristic of synthetic todorokite transformed from birnessite [75], Figure S5: FEG-SEM energy-dispersive spectrometer (EDS) image and Elemental maps of todorokite nanoplatelets. Image scale bar is 500 nm, Table S1: Powder X-ray diffraction mineralogy of various Ore/MnMISS types, Table S2: SEM-energy-dispersive X-ray spectroscopy (EDX) spot analyses of crystalline todorokite (Figure S4B), Values in weight %.

**Author Contributions:** Conceptualization, S.P.K., M.I. and E.C.F.; Formal analysis, S.P.K., M.I., J.E.R., H.G., J.N. and K.D.; Funding acquisition, S.P.K., M.I., E.C.F. and H.G.; Investigation, S.P.K., M.I., E.C.F., J.E.R., H.G. and K.D.; Methodology, S.P.K., M.I. and E.C.F.; Project administration, S.P.K., M.I. and E.C.F.; Resources, S.P.K., M.I. and E.C.F.; Validation, M.I.; Visualization, S.P.K., M.I., E.C.F. and J.E.R.; Writing—Original draft, S.P.K. and M.I.; Writing—Review and Editing, S.P.K., M.I., J.E.R. and J.N. All authors have read and agreed to the published version of the manuscript.

**Funding:** This work was supported by the National and Kapodistrian University of Athens, Special Account for Research Grants (SARG) (grant numbers 70/4/3373, 70/4/6425 to SPK), the Biotransformations of Trace elements in Aquatic Systems (BIOTRACS) (Grant agreement ID: 514262 to KD): FP6-MOBILITY/Marie Curie Host Fellowships-Early stage research training (EST) (grant number MEST-CT-2004-514262), the Society of Economic Geologists (SEG) Hugh Exton McKinstry fund to KD, the Swedish Research Council (grant number 2012–4364 and 2017–04129 to MI), the Danish National Research Foundation (grant number DNRF53), a Villum Investigator (Villum Fonden) Grant to Don Canfield (No. 16518). Ernest Chi Fru was funded by a European Research Council grant no. 336092. JN publishes with the permission of the Executive Director British Geological Survey (UKRI).

**Acknowledgments:** We thank Liane Benning for access to and assistance with FEG-TEM and FEG-SEM analysis at Leeds University. We also thank two anonymous reviewers for their helpful suggestions for revision.

**Conflicts of Interest:** The authors declare no conflict of interest.

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
