# Peer review of "Precipitation of Mn Oxides in Quaternary Microbially Induced Sedimentary Structures (MISS), Cape Vani Paleo-Hydrothermal Vent Field, Milos, Greece"

_minerals, doi:10.3390/min10060536_

Round 1
Reviewer 1 Report
see attached

Author Response
Response to Reviewer 1 Comments
The manuscript entitled “Precipitation of Mn oxides in Quaternary microbially induced
sedimentary structures (MISS), Cape Vani Paleo-Hydrothermal Vent Field, Milos, Greece”
describes a biological role in the formation of these Mn-rich deposits. The authors use a variety of analytical techniques, including SEM, carbon isotopes, biomarkers and EPR, to make a convincing case that microbes not only initiated the precipitation of Mn(III/IV) oxides, but as well played a key role in their diagenetic modification. Overall, I found this to be an interesting study that ultimately should be published. However, I do feel that the authors need to make some moderate revisions in the form of: (1) a thorough edit of the text to eliminate some odd phrasing, and (2) better describing the main objectives and findings of this study. In the case of the latter, the abstract highlights this point because the final sentence states: “the Cape Vani—type Mn(III/IV) oxide-mineralized microbe-siliciclastic sediment systems widen the spectrum of environments responsible for marine Mn biometallogenesis”. To be honest, this is too vague, and it underplays the significance of this work. A third point is a partial re-write of the discussion to make the text clearer and less ambiguous – at times it seems like there are contradictory statements about the significance of microbes and which microbes are playing a role.
Response: We are grateful for the time Reviewer 1 has taken to review our paper and for all the comments presented. The reviewer’s attention to detail has transformed both the quality of the complex interpretations, better bringing out the significance of our work. We will like to point out that all suggestions have been helpful. Below we address his critical comments point by point. We have also submitted the latest version of the manuscript, with the changes requested by the reviews in red.
We have better described the main objectives and findings of our study in the revised abstract, which now reads as:
Understanding microbial mediation in sediment-hosted Mn deposition has gained importance in low-temperature ore genesis research. Here we report Mn oxide ores dominated by todorokite, vernadite, hollandite, and manjiroite, which cement Quaternary microbially induced sedimentary structures (MISS) developed along bedding planes of shallow-marine to tidal-flat volcaniclastic sandstones/sandy tuffs, Cape Vani paleo-hydrothermal vent field, Milos, Greece. This work aims to decipher the link between biological Mn oxide formation, low-T hydrothermalism, and, growth and preservation of Mn-bearing MISS (MnMISS). Geobiological processes, identified by microtexture petrography, scanning and transmission electron microscopy, lipid biomarkers, bulk- and lipid-specific δ13Corganic composition, and field data, and, low-temperature hydrothermal venting of aqueous Mn2+ in sunlit shallow-waters, cooperatively enabled microbially-mediated Mn(II) oxidation and biomineralization. The MnMISS biomarker content and δ13Corg signatures, strongly resemble those of modern Mn-rich hydrothermal sediments, Milos coast. Biogenic and syngenetic Mn oxide precipitation established by Electron Paramagnetic Resonance (EPR) spectroscopy and petrography, combined with hydrothermal fluid flow-induced pre-burial curing/diagenesis, may account for today’s crystalline Mn oxide resource. Our data suggests that MISS are not unique to cyanobacteria mats. Furthermore, microbial mats inhabited by aerobic methanotrophs may have contributed significantly to the formation of the MnMISS, thus widening the spectrum of environments responsible for marine Mn biometallogenesis.
Specific comments
Point 1: L50: I don’t understand why the authors have Mn(II) <—> Mn(III/IV) and Mn(IV). Why is Mn(IV) repeated?
Response 1: We agree with the above notice. “{Mn(II) <—> Mn(III/IV) and Mn(IV)}”, has been changed to “{Mn(II) <—> Mn(III/IV)}”
Point 2: L71: Manganese fixation seems wrong because it sounds like a biological process (e.g., N2-fixation). Perhaps manganese accumulation or enrichment?
Response 2: We have rephrased the above unsuitable word as you advised. “Manganese fixation” is revised to “Manganese enrichment”.
Point 3: L87: Replace shallow euphotic ‘hydro- and bio-spheres’ with ‘environments’.
Response 3: We have rephrased those words as you advised.
“hydro- and bio-spheres” is revised to “environments”
Point 4: L106: What is a high enthalpy geothermal system?
Response 4: “…high enthalpy geothermal system” is revised to… “…high enthalpy geothermal system (i.e. conventionally exploitable at an average of 3–5MW electric power per well)…”
Point 5: Figure 1: At present the figures are too small. Consider placing ‘a’ on top of ‘b’.
Response 5: For clarity and better quality, we have replaced the old “a”, with three panels “a”, “b” and “c”, and changed the legend accordingly,
We have positioned new “a”, “b” and “c” on top of old “b” now “d”.
Point 6: L119: Remove “will deal only with’ with ‘describes’
Response 6: “will deal only with” is revised to “describes”
Point 7: L121: change to ‘wt.%’
Response 7: “14.6% Mn, 12.8% BaSO4, 62% SiO2, and 7.5% Fe2O3” is revised to “14.6 wt. % Mn, 12.8 wt. % BaSO4, 62 wt. % SiO2, and 7.5 wt. % Fe2O3.”
Point 8: L123-124: Briefly explain what negative Ce and positive Eu anomalies means.
Response 8: “…negative Ce anomalies and positive Eu anomalies [44].” is revised to: “…negative Ce anomalies and positive Eu anomalies, which characterize marine hydrothermal Mn deposits, and, the presence of K-feldspar and barite, respectively [41, 44].”
Point 9: L145: What is the significance of the quartz-barite feeder veins in terms of the MISS?
Response 9: “; moreover, they are closely associated with the quartz-barite feeder veins” was deleted form Line 145, and incorporated to the last sentence of the respective paragraph, which reads like this:
“Moreover, the Cape Vani MnMISS are closely associated with the quartz-barite feeder veins, signifying the link between MISS growth and low temperature seafloor hydrothermal fluid venting.”
Point 10: L152: rephrase ‘probably acted as Mn(II) supplier”.
Response 10: “probably acted as Mn(II) supplier” is revised as suggested to read as: “, which is probably a significant source of dissolved Mn(II)”.
Point 11: L155-156: How do you get an age from S and O isotopes?
Response 11: This has been explained in detail in pages 21-22 (please see 4.1.3. Interpretation of the S and O Isotopes in Barite) of the publication by Ivarsson et al (2019) [43]. {Ivarsson, M.; Kilias, S.P.; Broman, C.; Neubeck, A.; Drake, H.; Fru, E.C.; Bengtson, S.; Naden, J.; Detsi, K.; Whitehouse, M.J. Exceptional preservation of fungi as H2-bearing fluid inclusions in an early Quaternary paleohydrothermal system at Cape Vani, Milos, Greece. Minerals 2019, 9, 749; doi:10.3390/min9120749}
The text has been updated by adding a reference to read as: “Sulphur (δ34S) and oxygen (δ18O) isotope evidence has shown that the investigated MnMISS are younger than 1.5 Ma, and possibly formed about 0.5 Ma ago;” is revised to “Sulphur (δ34S) and oxygen (δ18O) isotope evidence has shown that the investigated MnMISS are younger than 1.5 Ma, and possibly formed about 0.5 Ma ago [43];”
Point 12: L158: Do you mean the fungal remains are preserved within fluid inclusions?
Response 12: The rationale for the phrase “Furthermore, fungal remains, exceptionally preserved as fluid inclusions, …… from feeder veins [43]” is detailed in the publication [43] (Ivarsson et al., 2019), which is cited in the text. See also Response 11.
Point 13: Section 2.2: Add a line here mentioning what FEG-SEM is used for.
Response 13: We have amended the text accordingly, by adding the following phrase:
“FEG-SEM imaging was used for characterizing the morphology and distribution of mineral phases in the samples”.
Point 14: L184: “… performed on powdered samples placed onto …”
Response 14: Revised as suggested to read:
“High-Resolution (HR)-TEM (HRTEM) investigations were performed on powdered samples placed onto a holey carbon support grid (Agar Scientific Ltd.) using a FEI CM200 FEG-TEM operating at 197 kV with a point resolution of ca. 2.3 AËš, and fitted with a Gatan Imaging filter (GIF 200), at the Institute for Materials Research, University of Leeds.”
Point 15: Section 2.3: Similarly, add a line explaining what EPR does. You do it in the discussion, but it would be better doing it here.
Response 15: The following phrase, has been added as advised to Section 2.3:
“Electron Paramagnetic Resonance (EPR) spectroscopy is a technique to study paramagnetic species in an applied magnetic field, and it is used here for analysing Mn oxides to indicate a biogenic signature (see Section: 4.2.1. Biogenicity)”.
Accordingly, the following phrase, has been deleted from Section 4.2.1.: “EPR is a technique to study paramagnetic species in an applied magnetic field.”
Point 16: L236: How do you know you have todorokite? You only mention XRD on p9.
Response 16: We agree. Accordingly the following phrase (L236-L237) has been deleted, and does not appear in the revised text: “Todorokite (see below) may contain appreciable concentrations of Ca, and Mg, and K.”
Point 17: L243: Add a line describing why this particular ternary plot is used. Some of us are not familiar with it and won’t necessarily understand why high Mn means diagenetic versus being at the Fe corner instead.
Response 17: A phrase has been added as suggested, and the text has been updated to:
A number of geochemical classifications have been created that discriminate among marine hydrogenetic, Fe-Mn crust/nodule, diagenetic and hydrothermal origins for Mn oxides [37, 68, 69]. Accordingly, the MnMISS samples cluster within the diagenetic and hydrothermal marine fields based on the Mn-Fe-(Ni+Cu+Co)*10 ternary diagram (Figure 3).
Point 18: L251: What is PXRD? You only mention it later on L318.
Response 18: “PXRD” is revised to “powder X-ray diffractometry (PXRD)”
Point 19: Table 1: Why are there several tables all labelled Table 1?
Response 19: Table 1 is lengthy and spans multiple pages, i.e. the same Table is broken into three pages, with additional captioning that is revised to “Continued Table 1.”
Point 20: L285: The authors state: “Mn minerals have replaced almost all detrital grains”. Do you mean cemented instead?
Response 20: We agree. ”replaced” is revised to “cemented” throughout the text.
Point 21: Figure 3: Add arrows for Mn cement in panel (A).
Response 21: White arrows have been added for Mn cement in panel (A). The legend of Figure 3 has been updated to include “White arrows for Mn cement”, and now reads as:
“Figure 3. SEM backscatter …………… (A) Highly mineralized MnMISS layer with Mn oxide cement─supported texture (left side) grading into grain supported mineralized layer having minor Mn-oxide cement (right side). White arrows for Mn cement. (B) Colloform layering…….……………….”
Point 22: L319-320: What does “for the first time” mean in this context? For this site?
Response 22: According to reviewer comment, we felt that it is better to delete “for the first time”, for clarity of the relevant phrase, which now reads as:
The presence of micro- and nano-crystalline todorokite cement has been revealed by means of PXRD, BSE-EDS, FEG-SEM and HR-TEM investigations.
Point 23: L325: What is “layer-to-tunnel” todorokite?
Point 24: L326-327: The line “…. occur at natural oxic-anoxic sediment interfaces through electron transfers between microbes, minerals, organics, and metals[5,74–78]” doesn’t really make sense.
Please reword.
Response 23-24: (“todorokite”) is intended as an attribute of ”tunnel”. We agree that the phrasing created difficulty in understanding the meaning of the sentence. We felt that it is better to delete “(todorokite)” for clarity. Moreover, the sentence, has been rephrased as suggested to:
Interestingly, the key driving factor for this layer-to-tunnel phase transformation of Mn oxides is redox reactions through electron transfers between microbes, minerals, organics, and metals; these reactions occur at natural oxic-anoxic interfaces, for instance those in marine sediments [5,74–78]
Point 25: Figure 4: Explain the difference between (OA) lateral oriented attachments vs. (P) plate-like morphology because they appear similar.
Response 25: The relevant text (underlined) of the Figure 4 legend of has been revised as suggested. The new text reads like this:
“Figure 4. High–resolution …….………….. (OA): acicular particles elongated up to 200 nm, which are laterally aggregating together, forming stable laths up to 100 nm wide; (P): plate-like morphology, with plates comprised of overlapping todorokite laths—………………………… spacings in the a direction.”
Point 26: L399: Again, why is this table listed as Table 1?
Response 26: ”Table 1” has been corrected to “Table 2”, in the caption and the text.
Point 27: Also, does the lack of organic carbon mean that this was a C-limited environment? But then, how does that fit with it being a mat?
Response 27: Please see Response 42 below
Point 28: Discussion: I think this section needs a re-write because after reading through it a couple of times, I am still unclear about the specific roles of cyanobacteria vs. methanotrophs in the formation of the Mn-MISS.
Response 28: Below we address all of the reviewer’s critical comments point by point, and the relevant text revisions are contained in the submitted annotated latest version of the manuscript, in red.
Point 29: L434-452: I found this paragraph problematic because it is not clear if the 5 bulleted points are there to provide a summary of what is to follow – which I presumed it would be, versus just listing 5 points of evidence that the Mn-MISS are biological.
Point 30: L435: Explain what was missing in ref 45 that it required reassessing.
Response 29-30: Points 27 and 28 above are collectively dealt with here.
We agree that this paragraph (lines 434-452), does not belong in its entirety to the preface of Section 4.1. “Biogenicity of the Cape Vani Microbially Induced Sedimentary Structures (MISS) revisited”, but to the introductory statements of “Section 4. Discussion”.
We have therefore revised, and removed, L434-L452 to Section 4, as second introductory paragraph, where it serves to provide a summary of what is to follow. The revised text reads as:
The previous approach for recognizing Cape Vani MISS [45] will be reassessed on the basis of newly established biogenicity criteria for identification of “proven MISS”, coined by Davies et al. (2016)[87] (see Section 4.1.). Subsequently, we will discuss Mn oxide deposition as the result of: (1) Direct activity of microorganisms, i.e. microbial oxidation of Mn(II) during MISS growth leading to “biomineralization”; (2) Synsedimentary/pre-burial diagenetic processes associated with MISS preservation. These refer either to Mn oxides being a by-product of organic matter produced by microbial degradation interacting with inorganic compounds (e.g. hydrothermal Mn) to form Mn oxide mineral precipitates during “organomineralization” [88], or replacement of decaying microbial organic matter by minerals as a component of “destructive biomineralization” [64] or “mat decay mineralization” [89] (see Section 4.2).
Furthermore, we have rewritten and restructured the preface of Section 4.1., also to include the requested explanation (Point 28). The new text reads as:
The investigated Mn–mineralized sedimentary structures have been earlier [46] characterized as MISS, based on biogenicity criteria reviewed in Noffke (2010)[64]. More recently, Davies et al. (2016)[87] noted that MISS should be treated with caution as they are a subset of sedimentary surface textures (SST) that comprise those of abiotic origin. They proposed a new classification scheme that categorises the degree of certainty in assigning SST to MISS, based on the weight of accessory evidence: i.e., Category B signifying definitively biogenic/microbial (proven MISS), and Category A definitively abiotic (proven abiotic); Category ab where there is uncertainty, and Category Ab or Category Ba where there is uncertainty but one interpretation is supported. Accordingly, the previous approach for recognizing MISS at Cape Vani [46] is reassessed below on the basis of all the biogenicity criteria coined by Davies et al. (2016)[87], which include previously missing ones: (1) Preserved organic material, i.e., lipids; and, (2) Geochemical biosignatures, i.e., bulk and lipid specific Corg isotope biomarkers. As detailed below, based on existing field [46] and new micro-textural and geochemical criteria (this study), the Cape Vani Mn-mineralized SST fulfil all the criteria for biogenicity in Davies et al. (2016) [87], and are therefore defined as Category B MISS.
Point 31: L462-483: Consider re-writing this paragraph using full sentences to explain the similarities to MISS. At present this paragraph is too dense, the punctuation is poor, and the italics are distracting. Are these supposed to be subheadings?
Response 31: This text has been revised as suggested. The new text reads like this:
4.1.1. Similarity of form to modern MISS
The Cape Vani MnMISS bear striking similarities to structures produced by modern microbial mats in their host sandy sediments [90,91], which include:
- Sedimentary features and structures related to growth, such as growth bedding (see Figure 5A in Kilias (2012)[46], and nodular to biscuit–like surface structures (see Figure 5C in Kilias (2012)[46])
- Structures generated from a change in growth direction, such as tufted microstructures (Figure 2E), macroscopic reticulated surface patterns resembling ‘elephant skin’ (see Figure 6 in Kilias (2012)[46]), and microscopic pseudocolumnar structures and microstromatolite─like fabrics of Mn oxides (Figure 2F). The tufted microstructures (Figure 2E) compare strikingly well in size and shape with tufted biofilm structures left by modern cyanobacterial mats in their host sediments [90–93], as well as experimental tufted cyanobacteria biofilms [94]. The tufted morphology suggests coordinated growth commonly known from cyanobacterial mats in modern environments [93]. Consequently, tufted microstructures may be interpreted as a result of vertical growth of phototactic cyanobacteria stabilized by EPS. This is supported by the Mn oxide─filled bubble─like structure (white arrow in Figure 2E) similar to oxygen-rich bubbles trapped within modern oxygenic cyanobacterial mat fabrics [93]
- Features and fabrics related to trapping and binding of sediment, such as microbial lamina-bound volcaniclastic detrital grains (Figure 2H,I)
- Features derived from physical mat destruction, such as desiccation and cracks with upturned and curled margins, and jelly rolls and cracks (see Figure 7 in Kilias (2012)[46]), gas bubbles aerating the jelly mat below supporting mat detachment (see Figure 9D in Kilias (2012)[46]), and erosional edges, chip production (see Figure 10 in Kilias (2012)[46])
- Features derived from mat decay: Punctured gas domes, indicating sealing by mats interacting with post-burial gas production (see Figures 9A, B in Kilias (2012)[46]).
Point 32: L497-499: Consider removing this unnecessary sentence.
Response 32: Lines 497 to 499 have been deleted.
Point 33: L504: Explain how you decided that the C isotopes are indicative of the Calvin cycle.
Response 33: In order to decide that the C isotopes are indicative of the Calvin cycle, we have compared the MnMISS bulk and lipid-specific δ13Corg signatures, to those extracted from the modern day Mn oxide-depositing mats, from Spathi Bay, SSE Milos coast, which are composed entirely of microbial guilds that fix and utilise carbon primarily by the Calvin cycle [86].
Accordingly, the old text below (underlined):
”Furthermore, bulk δ13Corg data suggest primary carbon fixation mainly associated with the Calvin cycle, the metabolic process that both drives carbon cycling and is ubiquitous in modern day hydrothermal sediments from Spathi Bay, Milos [86].”
has been revised to provide the requested explanation, and the new text reads as:
“The MnMISS display bulk and lipid-specific δ13Corg signatures, which are comparable to those extracted from the modern day Mn oxide (birnessite) depositing non-cyanobacterial mats (−18‰ to −20‰, and −19.7‰ to −22.6‰, respectively), in shallow-seafloor hydrothermal sediments from Spathi Bay, SSE Milos coast [86]. In addition, the modern mats, are composed entirely of microbial guilds that fix and utilise carbon primarily by the Calvin cycle [86]. Consequently, similarities in δ13Corg between the modern and past environments suggest primary carbon fixation mainly associated with the Calvin cycle in the MnMISS.”
Furthermore, please note, that immediately following the above, and in order to support our comparative approach, we have restructured and revised the text, and, added a new sentence (bold) regarding the modern mats; the revised text reads as:
Furthermore, the modern brown-capped sediments (Mn and Fe oxide-rich) share similar biomarkers with the MnMISS (Figure 5). Our lipid data strongly suggests that the system in which the MnMISS structures were created is similar to the processes that are currently occurring in the Spathi bay Mn-rich hydrothermal sediments. In addition, data (which will be made available in an independent communication) acquired by genetic fingerprinting of the Mn oxide-rich zones of the modern mats, using DNA specific markers, suggests that the modern Mn-precipitating mats are associated with Mn(II)-oxidizing Bacillus species related to lineages found in the Guaymas hydrothermal vent field, California [99], and widespread α-Proterobacterial Mn(II)-oxidizing genes [100].
Point 34: L508-509: What is the significance of this sentence? So, the Green River Shale also has hopanoids with a similar range, so what? Better to relate to experimental studies demonstrating that this range applies to methanotrophs.
Response 34: We agree and have therefore, replaced the sentence:
“A similar series and distribution of hopanoids have been reported in shales, oil deposits and crude oil [67,101], providing evidence for the degraded remains of a microbial community in this environment, rather than new contaminants”
by
“Experimental studies have demonstrated that a similar range of hopanoid δ13C values are biosynthesised by methanotrophs [103, 104], and similar fatty acid δ13C values have been found to originate from aerobic methanotrophs inhabiting shallow marine microbial mats [105]”. NOTE: 103, 104 and 105 are new references (Summons et al., (1994); Jahnke et al., (1995); Ding & Valentine, 2008.”
Point 35: L510-512: This sentence is confusing because it starts by stating that the hopanoids are unlikely from cyanobacteria but then ends by stating that cyanobacteria produced the O2 for MN(II) oxidation. This needs clarifying.
Response 35: We strongly agree that the sentence (L510 to L512) was not well-phrased, leading to the difficulty in understanding the meaning. It has now been revised to:
“The depleted δ13C values indicate cyanobacteria are an unlikely source for the hopanoids.”
Point 36: L512-514: Explain what the absence of those specific biomarkers means.
Response 36: The text has been updated by an additional sentence, to include the requested explanation, which reads as:
“The absence of those specific biomarkers may suggest that photorophic cyanobacteria are not a key component of the sampled MnMISS; however, biomarkers may have been actively recycled (possible in a low-carbon environment, Table 2) and re-metabolised by microorganisms.”
Point 37: L534-535: So, the authors are suggesting that the biological role in Mn(II) oxidation is via methanotrophs versus cyanobacteria. This is confusing because I thought the possible role of methanotrophs in the Mn cycle is via Mn(IV) reduction, not Mn(II) oxidation. Or, do the authors mean the Mn(II) was oxidized by other chemoautotrophs to Mn(IV) and later the methanotrophs reduced it back to Mn(II). Please clarify.
Response 37: To attempt to explain the biological role in Mn(II) oxidation via methanotrophs versus cyanobacteria, and how they are linked together, is a very complex issue and beyond the scope of the present study. Consequently, for clarity, we have deleted the following sentence and the reference cited:
“This suggests a similar ecosystem to the hydrothermal field systems described by Lein et al. (1997)[107] in the Manus and Lau basins where reduced Mn passed from the hydrothermal vent systems to be oxidized by chemoautotrophic and methanotrophic bacteria in oxygen rich ocean water.”
Moreover, we have added a new last paragraph to Section 4.1.3., which sums up what our data suggest, i.e. that cyanobacteria did not form Mn-MISS, and that methanotrophs may have contributed significantly to the formation of the MnMISS. The new paragraph reads like this:
“In summary, our data suggest that MISS are not unique to cyanobacteria mats, and methanotrophs may have contributed significantly to the formation of the MnMISS. This view is further supported by the ability of methanotrophs to form mats in seep systems [105, 111, 112]. As demonstrated, the extracted biomarker content of the ancient Mn mats strongly resemble those of the non-cyanobacterial Mn-depositing mats on the modern shallow-seafloor hydrothermal setting on the Milos coast (Fig. 5). These mats are predominated by genes associated with the Calvin cycle and display bulk and compound-specific δ13Corg signatures comparable to those extracted from the uplifted ancient MnMISS [86]. Similarly, the methanotrophs prominent in the modern mats [86] are experimentally known to produce hopanoids containing δ13C values in the range recorded in the present study [103, 104]. Furthermore, the data suggest that unlike the patchy appearance of the contemporary mats, resulting from weak and diffuse hydrothermal activity, similarly widespread mats on the Early Quaternary seafloor precipitated Mn released into seawater as a result of intense volcanic hydrothermal activity.”
Point 38: L537: Do the cyanobacteria produced methane? Surely you mean their biomass later supported the production of methane?
Response 38: Indeed, cyanobacteria produce methane. According to a very recent paper by Bizic et al. (2020), which we have cited ([109] in the revised latest version of the manuscript), it is suggested that the formation of methane by cyanobacteria contributes to methane accumulation in oxygen-saturated marine and limnic surface waters”.
Therefore, we have left our related sentence unchaged: “If, photoautotrophic cyanobacteria participated in the mat-constructing microbial group of the studied MnMISS, then these may have produced CH4 and sustained methanotrophy in the CVSB sunlit and oxic environment [110]. In any case, even if present, cyanobacteria are not a key component of the sampled MnMISS.”
Please see: Bižić, et al. (2020) Aquatic and terrestrial cyanobacteria produce methane. Sci. Adv. 2020, 6, eaax5343.
Point 39: L539: The authors suggest that cyanobacteria did not form Mn-MISS. If true, this seems an important finding because I thought most MISS were attributed to cyanobacterial mats? Do methanotrophs actually form mats?
Response 39:
In various aerobic and anaerobic environments methanotrophs can be the dominant members of mat communities in seep systems, please see:
Ding, H., and D. L. Valentine (2008), Methanotrophic bacteria occupy benthic microbial mats in shallow marine hydrocarbon seeps, Coal Oil Point, California, J. Geophys. Res., 113, G01015, doi:10.1029/2007JG000537
In the study by Ding & Valentine (2008) aerobic methanotrophs were studied in a shallow marine system. After both insitu and laboratory experiments, the authors found that aerobic methanotrophy had an important function in the white mats and that methanotrophs contributed up to 46% of the total fatty acids analysed. The δ13C values reported for fatty acids are in the same range as observed for hopanoids in our study.
Treude et al., (2007) Consumption of Methane and CO2 by Methanotrophic Microbial Mats from Gas Seeps of the Anoxic Black Sea. Applied and Environmental Microbiology.
They also co-occur with sulphide oxidizers in Marine volcano and hydrocarbon rich environments.
Paul et al., (2017). Methane-Oxidizing Bacteria Shunt Carbon to Microbial Mats at a Marine Hydrocarbon Seep. Frontiers in Microbiology.
The following sentence has been added to the revised text (see revised text above in Response 35), with the above references cited: ”This view is further supported by the ability of methanotrophs to form mats in seep systems [105, 111, 112]”
Point 40: L576-579: The authors need to explain what the various units (G, g) are.
Response 40: “G is the unit gauss and g is the (electronic) g factor (dimensionless)” has been added to the Fig. 6 legend
Point 41: L587-589: This sentence needs clarification because the authors have strongly argued for a biological role above but now seem to step back from it.
Response 41: According to the reviewer comment, we felt that it is better to delete lines 587-589: “Such values at the interface between biominerals and abiogenic Mn oxides are best explained as minor involvement of biology or a mix of biology and abiotic processes.”
Point 42: L614-617: Presumably you need O2 to form todorokite, so again what needs clarification is the role of the cyanobacteria vs. methanotrophs?Also, why is todorokite even preserved and not reduced via Mn(IV) reduction? I presume there was no methane formation then, so diagenesis stopped at aerobic respiration, but if so, what did the methanotrophs use?
Also, does the lack of organic carbon means this was a C-limited environment but how does that fit with it being a mat?
Response 42:
Based on our data, the overlying seawater column is implicated for the O2 supply, to form todorokite regarding the absence of phototrophic microbial lipid extracts and the more depleted hopanoid δ13C values, which indicate the presence of methanotrophic bacteria oxidizing methane aerobically [101,102], and do not support an in situ phototrophic mat source for marine photosynthetic O2. Consequently, the text has been updated by the addition of the the following sentence:
“The overlying seawater column is implicated for the O2 supply, to form todorokite; this is based on the absence of phototrophic microbial lipid extracts and the more depleted hopanoid δ13C values, which indicate methanotrophic bacteria oxidizing methane aerobically [101,102], and do not support an in situ phototrophic mat source for marine photosynthetic O2 (See Section 4.1.3)” The lack of organic carbon indicates this is a low carbon environment. However, hydrothermal systems are dynamic environments with high microbial functional diversity. Therefore, environmental variability may create a sedimentary environment with patchy density of mat and locally low carbon concentrations. It has been shown for modern hydrothermal soft marine sediments subjected to moderate hydrothermal venting that the processing of organic matter is strongly influenced by the production of chemosynthetic organic matter and that this is variable depending on the vent site [117].
“
Concerning the C-limited environment question, we have given an explanation that reads as:
The lack of organic carbon (Table 2) indicates this is a low carbon environment. However, hydrothermal systems are dynamic environments with high microbial functional diversity. Therefore, environmental variability may create a sedimentary environment with patchy density of mat and locally low carbon concentrations. It has been shown for modern hydrothermal soft marine sediments subjected to moderate hydrothermal venting that the processing of organic matter is strongly influenced by the production of chemosynthetic organic matter and that this is variable depending on the vent site [117].”
117 is a new reference. Bell et al., (2017) Hydrothermal activity, functional diversity and chemoautotrophy are major drivers of seafloor carbon cycling. Scientific Reports. 7.
Point 43: L636: The way this is described, i.e., Mn-MISS developed along bedding planes makes it seem like the Mn is secondary versus having formed at the sediment-water interface.
Response 43: According to reviewer comment, we felt that it is better to delete “and within beds”, which now reads as:
The MnMISS have been developed along bedding planes of Upper Pliocene to Lower Pleistocene shallow-marine/tidal-flat volcaniclastic sandstones/sandy tuffs closely associated with the Cape Vani paleo-hydrothermal vent field
----------------------------------------------------------------------------------------------------------------

Reviewer 2 Report
The manuscript provides a detailed look at the biogenicity of the unusual Mn mineralized sedimentary structures found in the Cape Vani paleo-hydrothermal vent field in Greece. The authors used geochemical and advanced microscopy techniques to characterize these Mn oxide ores, demonstrating to be biogenic in origin. The manuscript is an interesting contribution and should be published after minor revisions.
The goal of the manuscript is not well reflected in the abstract, as it is in the first paragraph of the discussion section. Please clearly identify the main goal of this work in the abstract section.
Why did the authors pre-treated the samples with HNO3, instead of HF or HCl, which are more appropriate treatments to remove inorganic matrices and concentrate the organic fraction?
Lines 512-513: “n” should be italicized.
Author Response
----------------------------------------------------------------------------------------------------------------
Response to Reviewer 2 Comments
The manuscript provides a detailed look at the biogenicity of the unusual Mn mineralized sedimentary structures found in the Cape Vani paleo-hydrothermal vent field in Greece. The authors used geochemical and advanced microscopy techniques to characterize these Mn oxide ores, demonstrating to be biogenic in origin. The manuscript is an interesting contribution and should be published after minor revisions.
We extend thankfulness to this reviewer for commenting on our manuscript. Below we
address his comments point by point. Below we address the comments point by point. We have also submitted the latest version of the manuscript, with the changes requested by the reviews in red.
Point 1: The goal of the manuscript is not well reflected in the abstract, as it is in the first paragraph of the discussion section. Please clearly identify the main goal of this work in the abstract section.
Response 1: This abstract has been reformatted as suggested to read as:
Understanding microbial mediation in sediment-hosted Mn deposition has gained importance in low-temperature ore genesis research. Here we report Mn oxide ores dominated by todorokite, vernadite, hollandite, and manjiroite, which cement Quaternary microbially induced sedimentary structures (MISS) developed along bedding planes of shallow-marine to tidal-flat volcaniclastic sandstones/sandy tuffs, Cape Vani paleo-hydrothermal vent field, Milos, Greece. This work aims to decipher the link between biological Mn oxide formation, low-T hydrothermalism, and, growth and preservation of Mn-bearing MISS (MnMISS). Geobiological processes, identified by microtexture petrography, scanning and transmission electron microscopy, lipid biomarkers, bulk- and lipid-specific δ13Corganic composition, and field data, and, low-temperature hydrothermal venting of aqueous Mn2+ in sunlit shallow-waters, cooperatively enabled microbially-mediated Mn(II) oxidation and biomineralization. The MnMISS biomarker content and δ13Corg signatures, strongly resemble those of modern Mn-rich hydrothermal sediments, Milos coast. Biogenic and syngenetic Mn oxide precipitation established by Electron Paramagnetic Resonance (EPR) spectroscopy and petrography, combined with hydrothermal fluid flow-induced pre-burial curing/diagenesis, may account for today’s crystalline Mn oxide resource. Our data suggests that MISS are not unique to cyanobacteria mats. Furthermore, microbial mats inhabited by aerobic methanotrophs may have contributed significantly to the formation of the MnMISS, thus widening the spectrum of environments responsible for marine Mn biometallogenesis.
Point 2: Why did the authors pre-treated the samples with HNO3, instead of HF or HCl, which are more appropriate treatments to remove inorganic matrices and concentrate the organic fraction?
Response 2: We strongly agree with the reviewer that this should definitely read “HCl”. “HNO3“ is a typo, therefore it has been deleted and replaced with “HCl”.
Point 3: Lines 512-513: “n” should be italicized.
Response 3: “n” has been replaced with “n”.
